# Data-Centric Human Preference with Rationales for Direct Preference Alignment

**Hoang Anh Just**
Virginia Tech
just@vt.edu

**Ming Jin**
Virginia Tech
jinming@vt.edu

**Anit Sahu**
Oracle
anit.sahu@gmail.com

**Huy Phan**
Amazon
huypq@amazon.co.uk

**Ruoxi Jia**
Virginia Tech
ruoxijia@vt.edu

## Abstract

Aligning language models with human preferences through reinforcement learning from human feedback is crucial for their safe and effective deployment. The human preference is typically represented through comparison where one response is chosen over another for a given prompt. However, standard preference datasets often lack explicit information on why a particular choice was made, presenting an ambiguity that can hinder efficient learning and robust alignment, especially given the high cost of acquiring extensive human annotations. While many studies focus on algorithmic improvements, this work adopts a data-centric perspective, exploring how to enhance learning from existing preference data. We propose augmenting standard preference pairs with rationales that explain the reasoning behind the human preference. Specifically, we introduce a simple and principled framework that leverages machine-generated rationales to enrich preference data for preference optimization algorithms. Our comprehensive analysis demonstrates that incorporating rationales improves learning efficiency. Extensive experiments reveal some advantages: rationale-augmented learning accelerates convergence and can achieve higher final model performance. Furthermore, this approach is versatile and compatible with various direct preference optimization algorithms. Our findings showcase the potential of thoughtful data design in preference learning, demonstrating that enriching existing datasets with explanatory rationales can help unlock improvements in model alignment and annotation efficiency.

## 1 Introduction

Human preference optimization is a critical stage in preparing large language models (LLMs) for real-world deployment. Its primary goal is to align model behavior with human expectations and prevent undesirable outputs (Christiano et al., 2017; Stiennon et al., 2020; Bakker et al., 2022). This alignment process typically relies on datasets containing prompts paired with ranked responses, which encapsulate human preferences. Popular methods like Reinforcement Learning from Human Feedback (RLHF)(Ouyang et al., 2022) leverage this data by first training a reward model to mimic human judgments and then using reinforcement learning(Schulman et al., 2017) to optimize the LLM policy against this reward signal. More recently, direct preference optimization (DPO) (Rafailov et al., 2024) and its variants offer an alternative approach, directly optimizing the policy using preference pairs through a loss function derived from an implicit reward model, thus avoiding the complexities of training an explicit reward model.

Despite their effectiveness, these methods face significant challenges. Models can overfit to the preference dataset (Azar et al., 2024), suffer performance degradation on broader

tasks (Pal et al., 2024), learn to exploit the rewards (Amodei et al., 2016), or generate excessively long and unhelpful responses (Park et al., 2024). Furthermore, acquiring high-quality human preference data is notoriously costly and time-consuming Zheng et al. (2023); Tan et al. (2024); Liu et al. (2024); Zhou et al. (2025). Yet, insufficient data can lead to underfitting, failing to achieve robust alignment Jinnai & Honda (2024). Much research has focused on mitigating these issues through algorithmic advancements, such as regularizing the optimization objective Pal et al. (2024); Amini et al. (2024); Park et al. (2024) or developing entirely new learning formulations Ethayarajh et al. (2024); Hong et al. (2024); Yuan et al. (2023); Munos et al. (2023); Swamy et al. (2024); Wu et al. (2024). While recent techniques such as SimPO (Meng et al., 2024) and SPPO (Wu et al., 2024) have demonstrated remarkable performance gains, often enabling smaller models to outperform larger ones on benchmarks like AlpacaEval Dubois et al. (2024), these improvements largely stem from modifications to the objective function or the generation of entirely new preference pairs.

However, human preference data are inherently valuable, representing the ground truth for alignment, and acquiring more is often prohibitively expensive. Generating synthetic data, while scalable, risks introducing biases or deviating from nuanced human intentions. This motivates a shift in focus: instead of solely pursuing algorithmic refinements or creating new datasets, can we extract more value from the existing, carefully curated human preference data? Our study adopts this data-centric perspective, specifically exploring how to enhance model alignment using direct preference alignment methods by improving the way models learn from the available human preference data. We pose the central question: How can we help the model better learn from existing human preference data to align more effectively with human expectations?

A key limitation of current preference datasets lies in their opacity regarding the reasons behind a preference. Given a prompt and two responses, why is one preferred over the other? While the distinction may be obvious in some cases (e.g., factual correctness), it becomes highly ambiguous when responses are closely matched in quality or exhibit subtle trade-offs. Superficial features like response length offer unreliable cues; longer responses might be favored for comprehensiveness in one context, while shorter ones are preferred for conciseness in another. Furthermore, human preferences can be subjective and stem from diverse, unstated reasoning. Without explicit explanations explaining these underlying criteria, the model faces a significant learning challenge. This ambiguity can lead to data inefficiency, requiring more examples to discern patterns, or worse, cause the model to learn spurious correlations, hindering true alignment and potentially degrading performance.

Figure 1: Comparison between the current pair-wise preference dataset used for preference learning and the enriched dataset with added rationales.

For the reasons outlined above, we propose a natural extension to the standard preference dataset structure by incorporating rationales. We define a rationale as an explanation accompanying each preference pair, explaining why the chosen response was preferred over the rejected one for the given prompt. This approach aims to provide the model with deeper contextual understanding during preference learning. This idea also draws inspiration from social studies (Mitchell et al., 1986; Chi et al., 1994; Crowley & Siegler, 1999; Williams & Lombrozo, 2010), showing that adding explanations to answers improves one's understanding of the problem compared to individuals who do not provide explanations.

We hypothesize that by enriching existing human preference datasets with rationales, models undergoing direct preference alignment can develop a more nuanced understanding of the underlying human values. This enhanced comprehension is expected to translate into improved preference modeling, higher performance on alignment benchmarks, and greater annotation efficiency, allowing models to learn human preferences more effectively from the available annotated data before potentially needing to resort to further costly data collection.

**Contributions.** This paper provides a new data-centric perspective on direct preference learning. We list the summary of contributions:

▮ We introduce rationales into the human preference learning framework, where rationales explain the reasons behind the preference for a particular response and derive a straightforward formulation for the preference function to extend the rationales and show how to adapt our method to current direct preference learning algorithms such as DPO.

▮ We analytically examine the impact of the rationale on preference training through the lens of information theory and empirically show the impact of preference learning with rationales.

In a broader context, our approach presents a new paradigm for data-centric research in language modeling: rather than focusing on pruning samples to distill the most informative pieces from a dataset Albalak et al. (2024), we explore how to enrich each sample's information content and examine its impact. The promising results presented in this paper demonstrate the effectiveness of enhancing individual samples' information content in preference learning and suggest that this approach may hold potential for improving learning in other domains.

## 2 Related Work (extended version in Appendix A)

Aligning large language models (LLMs) using techniques derived from human preferences is an important step for current LLMs (Casper et al., 2023; Christiano et al., 2017; Ziegler et al., 2019; Stiennon et al., 2020). While initial methods often relied on explicit reward modeling followed by reinforcement learning algorithms like Proximal Policy Optimization (PPO)(Schulman et al., 2017; Ouyang et al., 2022), newer approaches such as Direct Preference Optimization (DPO)(Rafailov et al., 2024) bypass explicit reward models. Despite their success, both PPO-based RLHF and DPO face challenges, such as overfitting to the preference data, underfitting due to data scarcity (Azar et al., 2024; Pal et al., 2024), and reward hacking (Amodei et al., 2016). To address these issues, researches introduce length-regularized DPO (R-DPO) (Park et al., 2024), odds ratio preference optimization (ORPO) (Hong et al., 2024), Kahneman-Tversky optimization (KTO) (Ethayarajh et al., 2024), or ranked responses methods (Yuan et al., 2023). General preference modeling, moving beyond Bradley-Terry assumptions (Bradley & Terry, 1952; Bertrand et al., 2023), offers alternatives such as Nash preference optimization (Munos et al., 2023; Swamy et al., 2024; Rosset et al., 2024; Wu et al., 2024). However, these methods focus primarily on algorithmic enhancements. A separate line of data-centric work aims to improve alignment by synthesizing entirely new preference datasets, often leveraging stronger models or self-critique mechanisms (Yang et al.; Pace et al.; Meng et al., 2024; Wu et al., 2024; Wang et al., 2024). Our work pursues a distinct, orthogonal approach. Instead of modifying the optimization algorithm or generating new preference pairs, we focus on enhancing the utility of existing preference datasets for direct optimization methods by introducing rationales that explain why one response was preferred. While related data-centric concepts which incorporate critiques to preference data have shown promise, particularly to refine reward models (Ye et al., 2024; Ankner et al., 2024; Yu et al., 2024), our rationales are designed for direct preference optimization methods and differ from critiques in design. Particularly, our rationales explain the difference between the responses, the critiques are made for each preference individually and independent of other preference. We provide an extended discussion of related work in the Appendix A.

## 3 Method

In this section, we introduce the incorporation of rationales into preference learning and show the derivation of adapting current methods. We present a demonstration of extending the direct preference optimization (DPO) algorithm to incorporate the rationales, while similar extensions can be applied to other variants of DPO. Further, we analyze theoretically the possible impact of rationales through the perspective of information theory.

### 3.1 Preliminaries

**Notations.** Let $\mathcal{D}$ denote the pair-wise preference dataset of size $N$, $\mathcal{D} = \{x^{(i)}, y_w^{(i)}, y_l^{(i)}\}_{i=1}^{N}$, where $x^{(i)}$ is a context, $y_w^{(i)}$ is the preferred/chosen/winning response to the context $x^{(i)}$ over the unpreferred/rejected/losing response $y_l^{(i)}$. Let $\pi_\theta$ and $\pi_{\text{ref}}$ denote the policy to be preference optimized and the reference policy respectively. In our setting, the policies are the language model to be preference trained and the base or supervised fine-tuned SFT model, respectively. To compute the joint probability of the autoregressive language model $\pi$ generating the response $y$ given the prompt $x$, we compute the product of probabilities after observing each token: $\pi(y|x) = \Pi_{t=0}^{|y|} \pi(y_t|x, y_{0:t})$.

**Reward Modeling with DPO.** In the RLHF process (Christiano et al., 2017; Ziegler et al., 2019; Stiennon et al., 2020; Bai et al., 2022; Ouyang et al., 2022; Rafailov et al., 2024), the goal is to align the language model towards human preferences. The preferences ranking from the dataset $\mathcal{D}$ is assumed to be sampled from the latent reward function $r(x, y)$ and the preference function is assumed to be generated by the Bradley-Terry model (Bradley & Terry, 1952): $p^*(y_w \succ y_l|x) = \sigma(r^*(x, y_w) - r^*(x, y_l))$, where $\sigma$ is the sigmoid function. The reward function then can be estimated by minimizing the log-likelihood of the following objective $\mathcal{L}(r, \mathcal{D}) = -\mathbb{E}_{(x, y_w, y_l) \sim \mathcal{D}} [\log \sigma(r(x, y_w) - r(x, y_l))]$. Then the next step is to tune the language model with the reward model as follows by maximizing the rewards and not diverging from the fixed reference model: $\max_{\pi_\theta} \mathbb{E}_{x \sim \mathcal{D}, y \sim \pi_\theta(y|x)} [r(x, y)] - \beta \mathbb{D}_{\text{KL}} [\pi_\theta(y|x) \| \pi_{\text{ref}}(y|x)]$, where $\beta$ is a hyperparameter measuring the divergence between two policies. Alternatively, with a reparametrization of the Bradley-Terry preference model (Rafailov et al., 2024), the preference function can be expressed in terms of policy $\pi^*$:

$$p^*(y_w \succ y_l|x) = \sigma \left( \beta \log \frac{\pi^*(y_w|x)}{\pi_{\text{ref}}(y_w|x)} - \beta \log \frac{\pi^*(y_l|x)}{\pi_{\text{ref}}(y_l|x)} \right). \tag{1}$$

Thus, to estimate the policy, Rafailov et al. (2024) proposes to directly minimize the log-likelihood of the following DPO loss: $\mathcal{L}(\pi_\theta; \pi_{\text{ref}}) = -\mathbb{E}_{(x, y_w, y_l) \sim \mathcal{D}} \left[ \log \sigma \left( \beta \log \frac{\pi_\theta(y_w|x)}{\pi_{\text{ref}}(y_w|x)} - \beta \log \frac{\pi_\theta(y_l|x)}{\pi_{\text{ref}}(y_l|x)} \right) \right]$.

### 3.2 Formulation of Preference Learning with Rationales

Although preferences are modeled given the preferred and unpreferred responses, there are nuances in the responses that are obscure for the model to comprehend and catch the differences between them. Therefore, our goal is to help the model learn preferences by providing guidance cues in the preference tuning process, which we call *rationales*. The rationales explain why a given response is preferred over the other response. For that reason, we extend the current preference learning with a data-centric technique to incorporate rationales, and we term this the *rationale-enriched preference function*, where the updated preference function is formulated as $p^*(y_w \succ y_l, r|x)$, and $r$ is the rationale from the updated dataset $\mathcal{D}' = \{x^{(i)}, y_w^{(i)}, y_l^{(i)}, r^{(i)}\}_{i=1}^{N}$. By the chain rule, we arrive at:

$$p^*(y_w \succ y_l, r|x) = p^*(y_w \succ y_l|x) \cdot p^*(r|x, y_w \succ y_l), \tag{2}$$

where $p^*(y_w \succ y_l|x)$ is the pair-wise preference term modeled in Section 3.1, and $p^*(r|x, y_w \succ y_l)$ is the probability of the rationale $r$ given the context $x$ and the preference $y_w \succ y_l$. Given the policy $\pi^*$, we can retrieve the probability of generating the rationale $r$ given the context $x$ and the preference $y_w \succ y_l$, $\pi^*(r|x, y_w \succ y_l)$. Similarly when retrieving

the probability of generating responses $y_w$ and $y_l$ for the prompt $x$, which are given in the preference dataset $\mathcal{D}'$, we can also retrieve the probability of generating rationale $r$ given $x, y_w$, and $y_l$, where $(x, y_w, y_l, r) \sim \mathcal{D}'$. In practice, we ask the policy language model $\pi_\theta$ to explain why the response $y_w$ is preferred over the response $y_l$ for the prompt $x$ and retrieve the probability of generating the rationale $r$. Thus, $p(r|x, y_w \succ y_l) = \pi_\theta(r|x, y_w \succ y_l)$.

**Adaptation to DPO Loss.** After deriving the rationale-enriched preference learning function, we extend the DPO method to incorporate rationales. By substituting Equation 1 and $p(r|x, y_w \succ y_l)$ into Equation 2, we can express the rationale-enriched preference function in terms of policy. We can then estimate the policy by optimizing $\pi_\theta$ through maximum likelihood using the following objective over the updated preference dataset $\mathcal{D}' = \{x^{(i)}, y_w^{(i)}, y_l^{(i)}, r^{(i)}\}_{i=1}^N$, which we term as rationale-DPO (RDPO):

$$\mathcal{L}_{\text{RDPO}}(\pi_\theta; \pi_{\text{ref}}) = -\mathbb{E}_{(x, y_w, y_l, r) \sim \mathcal{D}'} \left[ \log \sigma \left( \beta \log \frac{\pi_\theta(y_w|x)}{\pi_{\text{ref}}(y_w|x)} - \beta \log \frac{\pi_\theta(y_l|x)}{\pi_{\text{ref}}(y_l|x)} \right) + \gamma \log \pi_\theta(r|x, y_w \succ y_l) \right],$$

where $\gamma$ is the added hyperparameter for weighting the impact of rationales on the loss. We also provide experiments on adapting ORPO and SimPO with rationales in Section 4.

## 4 Evaluation

In this section, we evaluate the impact of rationales on direct preference learning. We conduct multiple experiments with two main goals in mind: **(1)** to understand how the added rationales affect the efficacy and efficiency of current preference learning algorithms, and **(2)** to determine the significance of rationale quality for effective learning.

### 4.1 Experimental Setup

**Datasets.** For our analysis, we focus on three preference datasets: Orca DPO Pairs (Intel, 2024), which is a pairwise preference dataset version of Orca (Mukherjee et al., 2023), a binarized UltraFeedback (Tunstall et al., 2023), which is a pair-wise version of UltraFeedack (Cui et al., 2023), and Anthropic Helpful and Harmless, which is a huamn preference dataset about helpfulness and harmlessness. For each dataset, we take 512 fixed samples as test set for winrate evaluations. We generate rationales and add rationales to the current datasets. We refer readers to Appendix C.9 for details on generating rationales. Given the test sets, we sample the responses from models trained with preference learning methods and compare the performance between the models by measuring the winrates between the corresponding responses.

**Models.** We investigate preference training on various large language models: Mistral-7B-v0.1, Mistral-7B-Instruct-v0.2 (Jiang et al., 2023), Zephyr-7B-Beta(Tunstall et al., 2023), and Llama3-8B-Instruct (AI@Meta, 2024). We use GPT-4o (Achiam et al., 2023) as a judge to evaluate the responses generated by the models and to retrieve the winrate scores. Please see Appendix C.11 for details. We provide full results with ablation on hyperparameters in Appendix C.2.

**Methods.** In our experiments, we study the integration of rationales into preference learning frameworks, such as DPO (Rafailov et al., 2024), which requires the SFT model for the reference model, and ORPO (Hong et al., 2024) and SimPO (Meng et al., 2024), which do not. To ensure fair comparison between DPO and RDPO, we fine-tune the base model with supervised fine-tuning (SFT) only using the chosen responses from the preference dataset. We extend the code implementation from human-aware loss functions (HALOs) repository (Ethayarajh et al., 2023) to adapt to our methodology and borrow the hyperparameters from each of the above methods in our study.

#### 4.1.1 Learning Dynamic with Rationales

**Versus SFT.** We carefully examine the dynamics of preference learning by studying how adding rationales to the current preference learning algorithms can impact performance. We compare the responses generated by the preference-aligned model against the responses generated by the SFT model and measure the winrates scores using the GPT-4o evaluator. To

study data annotation efficiency, we train the models on various training data sizes, ranging from 1,000 to 12,000 data points, for both RDPO and DPO training. We observe on the left side of Figure 2 that both models, DPO and RDPO models, achieve a better winning rate over the SFT model (more than 50%) on the Orca dataset with an increasing winning trend when training data increases. Additionally, we observe as the DPO model converges to around 60% winning rate at the 9,000 mark against the SFT model, the RDPO model achieves this rate at a smaller training data size with 3,000 training data points. Furthermore, we observe that the RDPO model can reach an even higher winning rate against the SFT than the DPO model, reaching above the 66% winning rate. We see a similar observation with the models trained on the UltraFeedback dataset on the right side of Figure 2. While RDPO can increase the computation time due to the addition of rationales, the model trained on rationales can converge earlier with fewer data points than DPO. This is especially important as the cost of collecting human preference data is high. Thus, improving annotation efficiency can potentially save further training costs. Additionally, with enough computation, RDPO can reach a better model than DPO does.

Moreover, we observe for the case of the UltraFeedback dataset, with more training data, the performance of the DPO model decreases. This can be attributed to the problem of DPO overfitting and exploiting length in longer responses Azar et al. (2024); Park et al. (2024). Indeed, the UltraFeedback dataset contains chosen responses that are longer on average (1,305 character length) than the rejected responses (1,124 character length). unlike the Orca dataset (784 and 978, respectively).

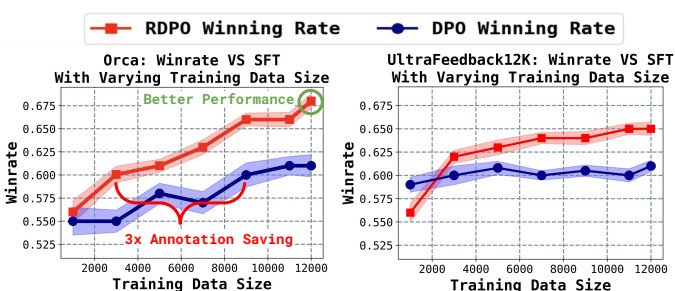

Figure 2: Winrate comparison between the models trained with RDPO and DPO. **Left:** Winrate against the SFT model trained on the Orca dataset. **Right:** Winrate against the SFT model trained on the Ultrafeedback dataset. X-axis denotes the training data size used for preference training of DPO and RDPO models.

**Versus DPO.** While we used SFT as a proxy to compare the performance between the DPO- and RDPO-trained models, here we directly compare the responses between these two models to measure the winrate for Orca and UltraFeedback datasets. We choose a DPO-trained model checkpoint for each dataset, where the winrate of the DPO model against the SFT model has converged, which is at 11,000 and 12,000, respectively. In Figure 3, we observe that the model trained with RDPO generates better responses on average than the model with DPO, even when trained with as little as 1,000 data points. With increasing training data, RDPO model improves the winrate to reach above 60% in both datasets. RDPO-trained model generates more of the preferred responses than the DPO-trained model does, even with $10\times$ fewer training points.

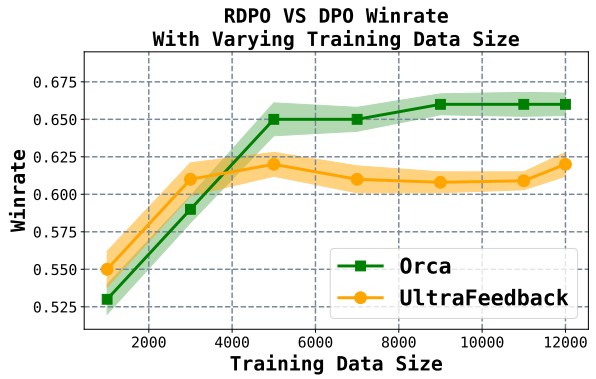

Figure 3: Winrate of RDPO model against the DPO model on respective datasets, Orca on the **left** and UltraFeedback on the **right**. The purple dashed line denotes the 0.5 mark.

**Adaptation to ORPO.** To demonstrate the flexibility of our rationale-enriched preference learning framework, we extend the ORPO preference learning algorithm Hong et al. (2024), which omits the SFT step, to include the rationales similar to the RDPO loss, and we call it RORPO. As shown in Table 1, rationales can enhance the performance of ORPO and achieve a better winrate over the vanilla ORPO trained model. By successfully adapting rationales to both ORPO and DPO, we emphasize the simplicity of the framework as well as the effectiveness of rationales in preference learning. We further study the adaptation of these methods with rationales and evaluate the preference-trained models on the instruction-following benchmark, AlpacaEval 2.0 (Li et al., 2023b; Dubois et al., 2024).

ORPO | 44  :  56 | RORPO

Table 1: Adapting rationales to the ORPO preference learning algorithm on Mistral-7B-v0.2-Instruct (Orca). Comparing the winrate of the ORPO- **(Left)** against the RORPO-trained model **(Right)**.

**Evaluation of the Impact of Rationales with AlpacaEval 2.0.** Here, we study the flexibility of our method by extending different preference learning methods, such as DPO (Rafailov et al., 2024) and ORPO (Hong et al., 2024), with rationales. In this experiment, we evaluate the performance of the trained models on the automatic instruction-following AlpacaEval 2.0 benchmark (Li et al., 2023b) with GPT-4-turbo as a judge and report the raw winrate, the length-controlled (LC) winrate (Dubois et al., 2024), which is robust against the verbosity bias that the raw winrate inherently entails, and the average response length. We train the instruction-tuned models, Mistral-7B-Instruct-v0.2 and Llama-3-8B-Instruct, on the Intel-DPO-Pairs dataset.

We observe in Figure 2 that DPO trained model improves the winrates on both models compared to the original model. Additionally, RDPO models further increase the win rates. We note that in the case of the Mistral model, the winrate decrease with ORPO preference training, which might be due to the lack of the reliance on the reference model, the behavior which is also observed in Ethayarajh et al.. Furthermore, after adding rationales, the (LC) win-

|          | Mistral-7B-Instruct-v0.2 | | | Llama-3.1-8B-Instruct | | |
|----------|-------|---------|--------|-------|---------|--------|
|          | LC WR | Winrate | Length | LC WR | Winrate | Length |
| Original | 17.11 | 14.72   | 1676   | 22.92 | 22.57   | 1899   |
| DPO      | 19.52 | 15.81   | 1632   | 26.02 | 22.52   | 1759   |
| RDPO     | 22.42 | 17.87   | 1627   | 27.55 | 23.69   | 1750   |
| ORPO     | 12.84 | 12.24   | 1782   | 23.11 | 20.27   | 1734   |
| RORPO    | 20.45 | 16.27   | 1618   | 26.55 | 23.45   | 1758   |

Table 2: The performance comparison of the original model, ORPO trained model, and DPO/ORPO with rationales (RDPO/RORPO) trained model on the AlpacaEval 2.0 benchmark.

rate not only increases but also surpasses of the original model. These results show the helpfulness of adding rationales into preference learning. Interestingly, we also note that rationale based models increase the winrates while their average response lengths are decreased compared to average response lengths of the original model, which is not a similar observation as seen in some current methods, such as SPPO (Wu et al., 2024) or SimPO (Meng et al., 2024).

**SimPO: Full Size Dataset** Here, we study the effectiveness of training with rationales on the full size dataset, and we adapt SimPO to incorporate rationales, and train the RSimPO model, accordingly. In particular, we train the LLaMA-3.1-8B-Instruct model on the UltraFeedback dataset and Anthropic HH dataset. After proper filtering, we are left with around 40K samples for each dataset. To show the effectiveness of RSimPO, we observe in Table 3 that RSimPO achieves a higher winrate

| LLaMA-3.1-8B-I  | UltraFeedack | | Anthropic HH | |
|-----------------|-----|-----|-----|-----|
| Dataset Size    | 20K | 40K | 20K | 40K |
| RSimPO Winrate  | 61  | 68  | 58  | 65  |

Table 3: The winrate of the RSimPO trained model on 20K and 40K dataset versus the SimPO trained model on the full 40K dataset.

against the SimPO trained model, achieving over 65% winrate on both datasets agains the SimPO model. Furthermore, to show the efficacy of RSimPO, we train the model on half of the dataset, and we still observe better performance compared to the fully trained SimPO model on 40K for both data sets.

## 4.2 Investigation of the Impact of Rationales

**Rationale-Only Optimization.** We study an the impact of the introduction of rationale term into the preference learning. To address this, we conducted a series of experiments to isolate the impact of each component and evaluate their combined effect. Specifically, we investigated an extreme case where the rationale loss alone drives preference optimization, with the DPO alignment loss set to zero. This approach was based on the hypothesis that rationales inherently encode preferences by combining preference-response pairs, the preferences themselves, and the associated reasoning processes, thereby providing a rich and effective training signal. For these experiments, we fine-tuned Mistral-7B-Instruct-v0.2 on the Orca dataset across three settings: RDPO (combining DPO and rationale loss), DPO (excluding rationale loss), and Rationale-Only (excluding DPO loss). The results, as shown in the Table 4, reveal that rationales alone can substantially improve model performance, achieving a high win rate of over 61% without explicit pairwise preference modeling. This improvement likely stems from the informational richness embedded in rationales, which compensates for the absence of pairwise alignment.

While DPO also demonstrated a majority win rate against the SFT baseline, training with both rationale and preference losses (RDPO) consistently achieves the highest win rate (64.5%) across both general and detailed settings. This highlights the benefit of integrating rationales into the preference objective, effectively leveraging the strengths of both losses to produce superior performance.

|  | RDPO | DPO | Rationale-Only |
|---|---|---|---|
| General | 64.5 | 59.1 | 61.8 |
| Detailed | 64.4 | 59.1 | 61.3 |

Table 4: The impact of different components of RDPO by measuring the win-rate of the target model against the SFT model. The results on Mistral-7B-v0.2-Instruct model using the Orca dataset.

These findings underscore the complementary nature of the rationale SFT loss and the pairwise alignment loss. While DPO explicitly optimizes reward margins, the rationale prediction loss provides supplementary supervision, enabling the model to learn the reasoning underlying response preferences. This integration not only strengthens the selection process but also accelerates training convergence. By combining these two approaches, RDPO amplifies their individual strengths, resulting in more efficient and effective preference learning.

| Generated By | RDPO vs DPO Winrate | |
|---|---|---|
| Mistral-7B-Instruct-v0.2 | 76-23 | 50-46 |
| Llama3-8B-Instruct | 73-27 | 52-45 |
| Phi3-Mini-4K | 75-25 | 51-49 |
| | Mistral-7B-Instruct-v0.2 | Llama3-8B-Instruct |
| | Trained On | |

Table 5: Studying the impact of different source of rationale generation (**Y-axis**) for the Orca dataset on the model training with RDPO (**X-axis**). Winrate of the RDPO model against the DPO model.

```
Given the prompt:
<prompt>
Which of the following assistant's responses is preferred and strictly follows
the prompt question? Why this assistant's response:
<chosen>
is preferred over the response provided below:
<rejected>
On a high and general level, why the response above is preferred over the
response below? Provide a general, high-level explanation for your reasoning
without going into the response's details.
```

Prompt 1: Prompt to the LLM to retrieve a general rationale.

| API Rationale Cost | $0.13 | $0.19 | $0.26 | $0.32 | $0.39 |
|---|---|---|---|---|---|
| Annotations Used | 1K | 1.5K | 2K | 2.5K | 3K |
| Annotations Saved | 3K | 6K | 6.5K | 6.8K | ¿10K |
| vs SFT Winrate | 54% | 56% | 58% | 60% | 62% |

Table 6: Cost-analysis breakdown of using RDPO with the OpenAI API (gpt-4o-mini model) to generate rationales, highlighting performance improvements relative to the number of annotations used and annotations saved.

**Rationale Source.** While collecting the human-annotated rationales could be high-quality, in practice, it is costly with time and resources. Therefore, we resolve to language models to generate rationales. In our experiments, we use the base models to create them. Here, we study the rationales coming from other sources and how they impact the RDPO training. We generate rationales for the Orca dataset on three different models: Mistral-7B-Instruct-v0.2, LLama3-8B-Instruct, and Phi3-Mini-4K Abdin et al. (2024). Then, we use these rationales to train the first two models. Results from Figure 5 show us consistent winrates against the DPO model with slightly better winrate from the same source as the base model. This shows that the rationales can be transferred to other models for preference training with rationales. Especially, leveraging models with small sizes $\{3, 7, 8\}$ billion parameters, we can generate rationales to improve preference learning using Prompt 1. However, we observe modest improvements on the Llama-3.1-8B-Instruct experiment, which can be attributed to two key factors. First, the inherent capability of Llama-3.1-8B-Instruct surpasses Mistral-7B-Instruct-v0.2, making substantial gains more challenging to achieve on this stronger baseline. Second, the general preference datasets like Orca and Ultrafeedback, which include pre-existing responses, may not be fully optimized for Llama-3.1-8B-Instruct.

**Quality of Rationales.** To investigate the quality of generated rationales, we incorporated the more capable LLM, GPT-4o, to evaluate the generated rationales in terms of helpfulness, relatedness, correctness, and coherence. In practice, we applied two primary criteria to ensure the high quality and reliability of the preference data used for training. We discarded data points where the preferred response was rated lower than or equal to the rejected response. This step was crucial for eliminating inherently inconsistent or ambiguous human judgments. We retained only instances where the preferred response received a score of 5 or higher on a 10-point scale. This ensured that the 'preferred' choice genuinely represented a positive example. We further examine the importance of the quality of rationales for preference learning in Appendix C.4, studying how different types of rationales would impact preference learning.

**Cost Analysis** In this section, we analyze the cost breakdown to assist project owners in evaluating trade-offs. Specifically, we present a cost-benefit analysis of the approach. The table below outlines the cost of using the API to generate rationales for a given number of annotations. It also highlights the RDPO win rate compared to the SFT model for each data budget and estimates the number of annotations potentially saved by using RDPO to achieve the same level of performance as DPO.

While open-weight models were used to generate the rationales in our study, the table illustrates the associated costs of utilizing an API model, specifically gpt-4o-mini. The pricing for this model is $0.150 per 1M input tokens and $0.600 per 1M output tokens. The results shown in Table 6 are based on the Mistral-7B-v0.2-Instruct model trained on the Orca dataset. We refer the reader to Appendix C.7 for more details.

## 5 A Simplified Theoretical Model for Rationales

To better understand rationale-enhanced preference learning, we employ machinaries from information theory to quantify benefits rationales provide in learning ground truth preferences under simplified assumptions. Formally, given query $X$, let the preference $Z$ be a binary random variable, with $Z = 1$ indicating that response $Y_1$ is preferred over response $Y_2$, and $Z = 0$ indicating the opposite. Let $R$ denote the rationale, $S = (X, Y_1, Y_2)$, and assume that the dataset $D = \{S_i, R_i, Z_i\}_{i=1}^n \sim \mu^n$ is sampled i.i.d. from a distribution $\mu$.

As an intuitive first step, we quantify the benefits using the conditional mutual information between the true preferences and rationale-implied preferences given the input-response pairs, i.e., $I(Z; g(R)|S)$, where $g(R)$ capture the preference inferred from the rationale $R$. Intuitively, this mutual information quantity characterizes the added value of rationales in understanding true preferences, measuring how much additional information rationales provide beyond what can be inferred from input-response pairs $S$. Our analysis demonstrates a closed-form relationship between rationale informativeness and its alignment with true preferences (see Appendix B.1 for detailed derivations of all results).

Next, we quantify the benefits provided by rationales by directly analyzing the generalization error for preference learning with and without rationales. Compared with analyzing the informativeness of rationales outlined above, this approach offers a more direct measure of the impact of rationales on learning outcomes. Note that the generalization error of preference learning still differs from the winrate metrics used in our experiemnts for evaluating the generation of preference-aligned models. However, they are intuitively connected under the DPO loss: the minimized test loss also indicates learning the optimal data generation policy that achieves the highest reward. We defer the detailed statement of the theorem and proof to Appendix B.2. In particular, our theoretical derivation shows that training with rationales can lead to improved lower generalization error when the rationale does not contain irrelevant information other than those predictive of the preference $Z$, and the learning process only captures the rationale information that is useful to predict $Z$. Despite being derived from a simplified model, our theoretical insights align well with experimental observations. For instance, our evaluation in Section C.4 demonstrates that detailed rationales, which may contain extraneous information, achieve lower sample efficiency compared to more general rationales that focus on preference-predictive content; furthermore, when we manually inject noise into rationales to misalign them with true preferences, we observe hampered preference learning, as indicated by lower winrates.

## 6 Conclusion & Limitations

In this work, we explored and validated a data-centric perspective on direct preference learning for aligning language models. Standard approaches train models on preference pairs (preferred vs. dispreferred responses), but often lack explicit information about why one response is superior, leading to ambiguity that can hinder learning efficiency and require substantial, costly datasets. To address this limitation and extract more value from existing human preference data, we introduced the concept of augmenting these pairs with rationales, which explain the reasoning behind each preference. We presented a straightforward framework for integrating readily available, machine-generated rationales into existing direct preference optimization pipelines. Our extensive empirical evaluations demonstrate that this rationale-augmented approach significantly enhances the learning process: it increase the effective use of each annotated data point, improves model convergence to higher performance levels, and helps mitigate undesirable behaviors. Furthermore, theoretical insights grounded in information theory corroborate the observed efficiency gains provided by rationales. Ultimately, our findings underscore the significant potential of thoughtfully designing the structure of preference data itself. Enhancing existing datasets with explanatory rationales offers a powerful and complementary strategy alongside algorithmic advancements for achieving more robust and efficient language model alignment. To date, we have integrated rationales into our training process and successfully trained models with up to 8 billion parameters. We encourage further investigation into the impact of rationales on preference learning, particularly exploring larger models and datasets. To facilitate research in this area, we make our code and datasets publicly available. With the development of unpaired preference learning algorithms, such as KTO Ethayarajh et al. (2024), it is important to extend the use of rationales to handle unpaired responses in future work, e.g., such as provided in the UltraFeedback dataset Cui et al. (2023) contains rationales for single responses without pairwise comparison. We also would like to study the impact of rationales on trained the reward models, which might have a different impact than the critiques from existing literature.

## Impact Statement

This paper presents a data-centric extension aimed at improving preference learning by enabling users to provide rationales for their choices, particularly in cases where such reasoning is not easily inferred by the model. By granting users greater control over data creation, this approach helps guide the model in effectively capturing user-specific preferences. Through responsible data curation and generation, this method has the potential to significantly influence model behavior in real-time applications.

## Acknowledgments

Ruoxi Jia and the ReDS lab acknowledge support through grants from the Amazon-Virginia Tech Initiative for Efficient and Robust Machine Learning, the Cisco Award, the Commonwealth Cyber Initiative Cybersecurity Research Award, the National Science Foundation under grants CNS-2424127, IIS-2312794, IIS-2313130, OAC-2239622, and OpenAI API research credits.

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

# Appendices

# A Extended Related Work

**RLHF with Reward Modeling.** Tuning large language models to align their outputs towards human preferences is crucial for controlling model behavior and maintaining desirable boundaries (Casper et al., 2023). To achieve this, RLHF has been introduced; aligning models through preference training (Christiano et al., 2017; Ziegler et al., 2019; Stiennon et al., 2020). Schulman et al. (2017) describe a method that typically involves two stages. The first stage learns a reward model using a preference dataset often modeled under the Bradley-Terry model Bradley & Terry (1952). The second stage fine-tunes the target policy model to maximize the rewards from the reward model, employing algorithms such as proximal policy optimization (PPO) proposed by Schulman et al. (2017) and adopted in Ouyang et al. (2022). A direct preference optimization (DPO) method that implicitly models the reward function was introduced by Rafailov et al. (2024). However, Azar et al. (2024) observe that RLHF and DPO are prone to overfitting due to the assumptions of the Bradley-Terry model. Conversely, Pal et al. (2024) explore the possibility of DPO underfitting when dealing with challenging responses that are difficult for the model to distinguish. Additionally, Park et al. (2024) note that DPO can exploit response length to maximize reward, proposing a length-regularized DPO (R-DPO) to address this issue. This should not be confused with our rationale-based DPO (RDPO) method. Interestingly, we observe that if rationales mention conciseness as a feature, then the length of responses is significantly reduced compared to SFT and DPO responses. The learning dynamics during preference tuning are analyzed in Im & Li (2024), emphasizing the importance of high-quality preference datasets for effective learning. They find that the more distinguishable the response pairs, the easier it is for the model to learn, leading to faster convergence. This has also been observed in Pal et al. (2024). However, designing such datasets is challenging, and it also remains important for models to learn from such nuanced responses, which appear in practice. We try to address the difficulty of model learning from intricate preferences by providing rationales during preference training. To improve efficiency over DPO, which requires an intermediate step to train the reference model, odds ratio preference optimization (ORPO) was introduced by Hong et al. (2024) to eliminate this step. Another method by Ethayarajh et al. (2024) adapts the Kahneman-Tversky human utility model to handle preference datasets with a single response (either chosen or rejected), removing the need for training the model on both responses. Conversely, Yuan et al. (2023) propose a preference method that considers multiple ranked responses for a prompt and optimizes over them. Our method can complement these methods by adding rationales into training. In this paper, we demonstrate an extension of our framework to ORPO.

**General Preference Modeling.** Reward modeling, however, can incentivize undesirable behaviors, such as "reward hacking" (Amodei et al., 2016), where agents maximize rewards without achieving the desired objective. Overfitting is another challenge, as exemplified in Azar et al. (2024). While effective for comparing two responses, the Bradley-Terry preference modeling relies on the assumption of transitivity, which may not hold true in practice (Bertrand et al., 2023). To address this, Munos et al. (2023) introduced general preference modeling, which directly learns general preferences by formulating a two-player, constant-sum game between policies. The goal is to maximize the probability of generating the preferred response against the opponent. The solution is the Nash equilibrium of this game, where payoffs are derived from the general preference function. Building upon this work, Munos et al. (2023) proposed an algorithm for the regularized general preference model, while Swamy et al. (2024) developed a solution for the unregularized formulation and introduced self-play preference optimization (SPO) as an iterative algorithm to reach the optimal solution. However, SPO suffers from data inefficiency due to its two-timescale update rules. To address this, Rosset et al. (2024) introduced an efficient direct Nash optimization (DNO) method that leverages the DPO formulation in practice. Additionally, Wu et al. (2024) proposed an efficient, scalable, iterative self-play method that generates responses generally preferred over others.

While previous efforts have introduced algorithmic enhancements for preference tuning, they have been limited to the existing framework of preference datasets with prompts and ranked responses. In contrast, our work is first to introduce rationales, a data-centric solution, into preference learning.

**Learning with Rationales.** The supervised learning framework typically involves training a model to learn the ground truth label for a given prompt without providing explicit explanations for the associations, which can lead to the model learning incorrect cues. To mitigate this issue, rationales have been integrated into the framework, offering explanations for the given associations. These rationales initially were generated by humans (Zaidan et al., 2007; Ross et al., 2017; Hase & Bansal, 2021; Pruthi et al., 2022). However, due to the high cost of human labor and the development of more capable large language models, rationales are now often automatically generated by these models, reducing the need for human involvement (Wei et al., 2022; Kojima et al., 2022). Given rationales, they have been used as guiding aids by incorporating them directly into the prompt during the training phase (Rajani et al., 2019; Zelikman et al., 2022; Huang et al., 2023) or at the inference stage (Wei et al., 2022; Kojima et al., 2022; Wang et al., 2022). Besides using them as additional context within the prompt, rationales can also serve as labels to train models to generate such explanations for their predictions (Wiegreffe et al., 2021; Narang et al., 2020; Eisenstein et al.; Wang et al.; Ho et al., 2023; Magister et al., 2022; Li et al., 2023a). In similar manner, rationales have been applied in knowledge distillation, where they are generated by a more capable models to supervise weaker models (Hsieh et al., 2023; Chen et al., 2024). In parallel with these advancements, we introduce rationales into the preference learning landscape, where rationales are used to explain the preference of one answer over another. Our findings demonstrate the effectiveness of rationales in preference learning, even when generated by the same model or a smaller-sized model.

**Synthetic Preference Data Generation.** Synthetic preference data generation plays a pivotal role in preference learning by creating new annotated datasets that capture user choices or preferences Yang et al.; Pace et al.; Meng et al. (2024). These methods focus on producing preference pairs that can serve as training data for models, enabling the exploration of diverse scenarios and reducing reliance on costly manual annotations. Furthermore, Wu et al. (2024); Wang et al. (2024); Meng et al. (2024) try to further synthesize the preference examples by iteratively generating new response pairs in on-policy manner. However, our approach diverges fundamentally from this objective. While synthetic data generation targets the creation of new datasets, our work emphasizes enhancing existing, fixed datasets with provided preferences pairs by incorporating rationales, thereby enriching preference annotations with explanatory depth. This distinction highlights the complementary nature of these methods: data generation addresses the early stage of creating foundational datasets, whereas rationale augmentation enhances the interpretability and utility of existing data. Attempting to compare these approaches directly would obscure their unique contributions. Instead, their synergy lies in how synthetic data generation can produce the raw preference pairs that are later refined through rationale augmentation, advancing the overall preference learning pipeline. Exploring how these two approaches can be combined to create synthetic datasets enriched with rationales is an exciting direction for future work, holding promise to further enhance the capabilities and generalizability of preference learning models.

# B  Theoretical Derivations

We begin with defining several standard quantities to be used throughout this section.

**Definition B.1.** Let $X, Y$ and $Z$ be arbitrary random variables, and let $D_{KL}$ denote the KL divergence. We denote $P_X$ as the marginal probability distribution of $X$, and $P_{Y|X}$ as the conditional distribution.

The entropy of $X$ is given by:

$$H(X) = -\sum_x P(X = x) \log P(X = x).$$

If $X$ is a binary variable with $p = P(X = 1) = 1 - P(X = 0)$, then we use $H(p)$ for $H(X)$.

The joint entropy of two random variables, $H(X, Y)$, is the entropy of their joint distribution.

The conditional entropy of $X$ given $Y$, $H(X|Y)$, is:

$$H(X|Y) = H(X, Y) - H(Y).$$

The mutual information between $X$ and $Y$ is:

$$I(X; Y) = D_{KL}(P_{X,Y} \| P_X P_Y).$$

The disintegrated mutual information between $X$ and $Y$ given $Z$ is:

$$I^Z(X; Y) = D_{KL}(P_{X,Y|Z} \| P_{X|Z} P_{Y|Z}).$$

The corresponding conditional mutual information is given by:

$$I(X; Y|Z) = \mathbb{E}_Z[I^Z(X; Y)].$$

If all entropies involved are finite, it can be shown that $I(X; Y) = H(Y) - H(Y|X)$.

## B.1  Mutual information analysis

Formally, given query $X$, let the preference $Z$ is a binary random variable, with $Z = 1$ indicating that response $Y1$ is preferred over response $Y2$, and $Z = 0$ indicating the opposite. Assume that the rationale-implied preference $R$ is a binary random variable, with $R = 1$ indicating a rationale that supports $Y1$ being preferred, and $R = 0$ otherwise. For example, if the rationale mentions that $Y1$ is more concise and informative than $Y2$, then $R = 1$. However, there can be cases where $R \neq Z$, as the rationale may not always align perfectly with the actual preference.

For the analysis, we consider the following model: 1) The rationale $R$ depends on the query-response (QR) pair $S = (X, Y1, Y2)$ and the preference $Z$, and is characterized by parameters $\beta$ and $\alpha$, where: $\beta = P(R = 1|Z = 1, S) = P(R = 0|Z = 0, S)$ represents the precision rate of consistency, and $\alpha = P(R = 1|Z = 0, S) = P(R = 0|Z = 1, S)$ represents the recall error due to inconsistency. 2) The preference $Z$ is modeled as $P(Z = 1|S) = f(S) + \epsilon$, where $f(S)$ captures the preference based on the observable query-response pair $S$, and the additive noise term $\epsilon$ is a simple way to account for unobserved factors influencing the complex human preference that are not captured in $S$. *The term $\epsilon$ is referred to as "bias" in the **main text** that accounts for the difference between the true preference and prediction based on the query and responses alone.*

**Theorem B.2.** *Under the given assumptions, the mutual information $I(Z; R|S)$ is given by:*

$$
\begin{aligned}
H(p + \epsilon) &- (\beta(p + \epsilon) + \alpha(1 - p - \epsilon)) \cdot H\left(\frac{\beta(p + \epsilon)}{\beta(p + \epsilon) + \alpha(1 - p - \epsilon)}\right) \\
&- (1 - (\beta(p + \epsilon) + \alpha(1 - p - \epsilon))) \cdot H\left(\frac{\alpha(p + \epsilon)}{\alpha(p + \epsilon) + \beta(1 - p - \epsilon)}\right),
\end{aligned}
$$

*where $p = f(S)$. The mutual information $I(Z; R|S)$ satisfies the following properties in three distinct regimes:*

*1. Uninformative rationale regime: If $\beta = \alpha = 0.5$, then $I(Z; R|S) = 0$.*

*2. Maximally informative rationale regime: If $\beta = 1$ and $\alpha = 0$, then $I(Z; R|S) = H(p + \epsilon)$.*

*3. Moderately informative rationale regime: If $\beta = 0.5 + \gamma$ and $\alpha = 0.5 - \gamma$, where $0 < \gamma < 0.5$, then $I(Z; R|S)$ increases with $\gamma$, ranging from 0 when $\gamma = 0$ (uninformative rationale) to $H(p + \epsilon)$ when $\gamma = 0.5$ (maximally informative rationale).*

The theorem highlights that the potential benefits of including rationales in the training process for preference learning tasks may vary in different regimes.

**Regime 1: Highly informative rationale ($\beta = 1$ and $\alpha = 0$):** In this case, the mutual information is solely determined by the entropy of the preference prediction from the query and responses, $H(f(S) + \epsilon)$. Let us interpret $f(S)$ as the query-response-dependent (QR-dependent) confidence generator that only depends on $S$, and $\epsilon$ captures the idiosyncrasies such as unknown confounding factors that influence the probability of preference. If the true probability $P(Z = 1|S) = f(S) + \epsilon$ is less than 0.5, i.e., $Z$ is most likely to be 0, a positive $\epsilon > 0$ means that the true probability $P(Z = 1|S)$ is less extreme than the confidence score $f(S)$ as it gets closer to 0.5 with increasing $\epsilon$. On the contrary, a negative $\epsilon < 0$ means that the true probability $P(Z = 1|S)$ is more extreme than the QR-dependent confidence $f(S)$ as it gets closer to 0 with increasing $|\epsilon|$.

From the perspective of the QR-dependent confidence generator $f(S)$ that tries to explain the preference based on QR pairs, a positive $\epsilon > 0$ would make it look more confident than it should be, i.e., overconfident, while a negative $\epsilon < 0$ would make it less confident (or more conservative) based on the QR sequence than it should be.

If it is likely to be overconfident based on the QR pairs, i.e., $\epsilon > 0$, then the more positive $\epsilon$ is, the more risk there is of being overconfident in the QR-dependent predictor $f(S)$. In this case, there is a lot of mutual information in $I(Z; R|S)$, so having rationales can "soften" the potential overconfidence by bringing additional information other than the QR pair, which would otherwise occur in QR pairs alone, as in traditional reward modeling. Similar analysis holds when $P(Z = 1|S) = f(S) + \epsilon$ is greater than 0.5, i.e., $Z$ is most likely to be 1.

*Key message:* Rationales are most useful when the reward modeling based on QR alone tends to have bias (i.e., overconfident).

**Regime 2: Uninformative rationale ($\beta = \alpha = 0.5$):** In this regime, the rationale provides no *additional* information about the preference, and the mutual information $I(Z; R|S)$ is zero.

**Regime 3: Moderately informative rationale (high precision $\beta = 0.5 + \gamma$ and low recall error $\alpha = 0.5 - \gamma$, where $0 < \gamma < 0.5$):** In this regime, it can be shown based on dereivative analysis that as $\gamma$ increases (more informative rationale), the terms involving $\gamma$ in the numerators and denominators of the conditional entropies become more prominent. The mutual information will increase with $\gamma$, as the rationale becomes more informative about the preference.

## B.2   Theorem B.3: Generalization Bound

Next, we analyze the sample complexity of training language models with and without rationales to predict preferences. We consider two regimes: *1)* **Training with rationale**: Let $\theta_{\mathrm{ra}} = \mathcal{A}_{\mathrm{ra}}(D) \sim P_{\theta_{\mathrm{ra}}|D}$ denote the parameters of the language model trained to predict $Z$ given $S$ and $R$. *2)* **Training without rationale**: Let $\theta_{\mathrm{un}} = \mathcal{A}_{\mathrm{un}}(D_{\backslash R}) \sim P_{\theta_{\mathrm{un}}|D_{\backslash R}}$ denote the output parameters trained to predict $Z$ given only $S$, where $D_{\backslash R}$ is a dataset $D$ with rationales removed. Given a loss function $\ell$ that measures the prediction of preference $Z$, the (mean) generalization error is $\mathrm{gen}(\mu, \mathcal{A}) = \mathbb{E}_{D,\theta\sim\mathcal{A}(D)}|\mathbb{E}_\mu[\ell(\theta)] - \mathbb{E}_D[\ell(\theta)]|$, where $\mathbb{E}_\mu[\ell(\theta)]$ is the expected loss on the true distribution (true risk) and $\mathbb{E}_D[\ell(\theta)]$ is the empirical risk.

We introduce the following conditions on the relationship between $S$, $R$, and $Z$, and the learning process: *1)* $H(R|Z) \le \eta_1$ and $H(Z|R) \le \eta_2$, i.e., the rationale $R$ is informative about

$Z$ (small $\eta_2$) without excessive irrelevance (small $\eta_1$). *2)* $I(\theta_{\mathrm{ra}}; S|Z, R) \leq \delta$ for some small positive constant $\delta$, i.e., the learned model $\theta_{\mathrm{ra}}$ does not capture much additional information from $S$ beyond what $Z$ and $R$ already provide. Condition 1 is supported by an effective procedure to generate useful rationale $R$. To justify Condition 2, if the learning algorithm is designed to focus on capturing the information in $R$, which is highly informative about $Z$ (per Condition 1 for small $\eta_2$), we can show that the model $\theta_{\mathrm{ra}}$ can accurately predict $Z$ without *needing* to capture much additional information from $S$ beyond what is already present in $Z$ and $R$. We provide rigorous but partial justification in Appendix B.3.

**Theorem B.3** (Generalization bounds). *Suppose the loss function $\ell$ is $\sigma$-subgaussian under the true data distribution. Under conditions 1 and 2, we have:*

$$\mathrm{gen}(\mu, \mathcal{A}_{ra}) \leq \sqrt{\frac{2\sigma^2}{n} \cdot (I(\theta_{ra}; Z) + \delta + \eta_1)}, \tag{3}$$

$$\mathrm{gen}(\mu, \mathcal{A}_{un}) \leq \sqrt{\frac{2\sigma^2}{n} \cdot (I(\theta_{un}; Z) + I(\theta_{un}; S|Z))}. \tag{4}$$

The proof relies on the mutual information-based generalization bounds (Russo & Zou, 2016; Xu & Raginsky, 2017) and the decomposition of the mutual information terms for both training regimes using the chain rule (see Appendix B.2). The terms $I(\theta_{\mathrm{ra}}; Z)$ and $I(\theta_{\mathrm{un}}; Z)$ can be expected to be similar as long as both regimes achieve good prediction of $Z$. Under the conditions of the theorem, we can observe that the sample complexity reduction depends on the gap between $I(\theta_{\mathrm{un}}; S|Z)$ and $\delta + \eta_1$; training with rationales can lead to improved sample efficiency when the rationale does not contain irrelevant information other than those predictive of the preference $Z$, i.e., $\eta_1$ is small, and the learning process only captures the rationale information that is useful to predict $Z$. The theoretical insights are supported by our experimental results. For instance, our evaluation in Section C.4 demonstrates that a detailed rationale achieves lower sample efficiency compared to more general rationales, which contain less irrelevant information beyond what is predictive of the preference; furthermore, we showed that irrelevant rationales, i.e., a large value of $\eta_1$, indeed hamper learning.

For the proof, recall that given a loss function $\ell$, the (mean) generalization error is $\mathrm{gen}(\mu, \mathcal{A}) = \mathbb{E}_{D, \theta \sim \mathcal{A}(D)} |\mathbb{E}_\mu[\ell(\theta)] - \mathbb{E}_D[\ell(\theta)]|$, where $\mathbb{E}_\mu[\ell(\theta)]$ is the expected loss on the true distribution (true risk) and $\mathbb{E}_D[\ell(\theta)]$ is the empirical risk. For fair comparison between $\mathrm{gen}(\mu, \mathcal{A}_{\mathrm{un}})$ and $\mathrm{gen}(\mu, \mathcal{A})_{\mathrm{ra}}$, some technical nuances arise. The key difference from the typical setup in Xu & Raginsky (2017) is that the true data distribution $\mu$ includes the distribution for the rationale $R$, but the training regime $\mathrm{gen}(\mu, \mathcal{A}_{\mathrm{un}})$ does not explictly use this information. However, it may seem unclear initially whether we should include that in the generalization bound, since $R$ is indeed generated based on $Z$, corresponding to the true Markov chain: $S \to Z \to R$.

To clarify, the Markov chain for the training without rationale is: $S \to Z \to R$, with additional arrows $Z \to \theta_{\mathrm{un}}$ and $S \to \theta_{\mathrm{un}}$, but no arrow from $R$ to $\theta_{\mathrm{un}}$.

Intuitively, we should account for this difference by arguing that, conditioned on the preference $Z$, the learned model $\theta_{\mathrm{un}}$ is conditionally independent of $R$. However, due to this difference, it seems prudent to reason from first principles.

Let's start by choosing the distributions $P$ and $Q$ for the Donsker-Varadhan variational representation of the KL divergence. We set $P = P_{S, R, Z, \theta_{\mathrm{un}}}$ and $Q = \mu^n \otimes P_{\theta_{\mathrm{un}}}$, where $\mu$ is the distribution for $(S, R, Z)$. Then, for any measurable function $f$, we have:

$$D(P\|Q) \geq \mathbb{E}_P[f(S, R, Z, \theta)] - \log \mathbb{E}_{(\bar{D}, \bar{R}, \bar{Z}, \bar{\theta}) \sim Q}[e^{f(\bar{S}, \bar{R}, \bar{Z}, \bar{\theta})}]. \tag{5}$$

Now, choose $f(S, R, Z, \theta) = \lambda(\ell_D(\theta) - \ell_\mu(\theta))$ for some $\lambda \in \mathbb{R}$, where $\ell_D(\theta)$ is the empirical loss on the dataset $D$ and $\ell_\mu(\theta)$ is the expected loss under the true distribution $\mu$. Substituting this into (5), we get:

$$D(P\|Q) \geq \lambda(\mathbb{E}[\ell_D(\theta)] - \mathbb{E}[\ell_\mu(\theta)]) - \log \mathbb{E}_{(\bar{D}, \bar{R}, \bar{Z}, \bar{\theta}) \sim Q}[e^{\lambda(\ell_{\bar{D}}(\bar{\theta}) - \ell_\mu(\bar{\theta}))}]$$

$$\geq \lambda(\mathbb{E}[\ell_D(\theta)] - \mathbb{E}[\ell_\mu(\theta)]) - \frac{\lambda^2 \sigma^2}{2n}, \tag{6}$$

where the second inequality follows from the fact that $\ell_{\bar{D}}(\bar{\theta})$ is $\sigma/\sqrt{n}$-subgaussian under $Q$, due to the subgaussian assumption on the loss function.

As (6) holds for any $\lambda \in \mathbb{R}$, it must also hold for the $\lambda$ that minimizes the right-hand side. This minimum occurs at $\lambda^* = n(\mathbb{E}[\ell_D(\theta)] - \mathbb{E}[\ell_\mu(\theta)])/\sigma^2$, yielding:

$$D(P\|Q) \geq \frac{n}{2\sigma^2}(\mathbb{E}[\ell_D(\theta)] - \mathbb{E}[\ell_\mu(\theta)])^2.$$

Taking the square root of both sides, we have:

$$\text{gen}(\mu, \mathcal{A}_{\text{un}}) \leq \sqrt{\frac{2\sigma^2}{n}D(P\|Q)}.$$

The key observation here is that $P_{S,R,Z,\theta_{\text{un}}} = P_{\theta_{\text{un}}|S,R,Z}P_{S,R,Z} = P_{\theta_{\text{un}}|S,Z}P_{S,R,Z}$, since $\theta_{\text{un}}$ is conditionally independent of $R$ given $S$ and $Z$ in this training regime. Therefore,

$$\begin{aligned}
D_{\text{KL}}(P\|Q) &= D_{\text{KL}}(P_{S,R,Z,\theta_{\text{un}}}\|\mu^n \otimes P_{\theta_{\text{un}}}) \\
&= D_{\text{KL}}(P_{\theta_{\text{un}}|S,Z}P_{S,Z}\|\mu^n_{\backslash R} \otimes P_{\theta_{\text{un}}}) \\
&= I(S, Z; \theta_{\text{un}}),
\end{aligned}$$

where $\mu_{\backslash R}$ denotes the marginal distribution of $\mu$ over $S$ and $Z$ (i.e., excluding $R$).

Hence, we have:

$$\text{gen}(\mu, \mathcal{A}_{\text{un}}) \leq \sqrt{\frac{2\sigma^2}{n} \cdot I(S, Z; \theta_{\text{un}})}.$$

Note that we use $I(S, Z; \theta_{\text{un}})$ instead of $I(S, R, Z; \theta_{\text{un}})$, which is the key difference from the typical setup in Xu & Raginsky (2017), where the learned model is assumed to depend on all the data.

We can then arrive at the result for training without rationale by noting:

$$I(\theta_{\text{un}}; S, Z) = I(\theta_{\text{un}}; Z) + I(\theta_{\text{un}}; S|Z).$$

For training with rationale, we have:

$$\begin{aligned}
I(\theta_{\text{ra}}; S, R, Z) &= I(\theta_{\text{ra}}; Z) + I(\theta_{\text{ra}}; R|Z) + I(\theta_{\text{ra}}; S|Z, R) \\
&\leq I(\theta_{\text{ra}}; Z) + H(R|Z) + I(\theta_{\text{ra}}; S|Z, R) \\
&\leq I(\theta_{\text{ra}}; Z) + \eta_1 + \delta,
\end{aligned}$$

where the first inequality is due to $I(\theta_{\text{ra}}; R|Z) \leq H(R|Z)$, and the second inequality follows from conditions 1 and 2. We can now apply [Thm. 1]Xu & Raginsky (2017) to yield the result.

## B.3 Supporting Lemmas

**Lemma B.4.** *Let $\hat{Z}$ be the estimate of $Z$ based on $\theta$. Let $P_e = P(Z \neq \hat{Z}|\hat{Z}, \theta)$ be the probability of error in predicting $Z$ using $\theta$. Suppose $P_e \leq \epsilon$. Then, we have that:*

$$P_e \geq \frac{H(Z) - I(R; \theta) - I(Z; \theta|R) - H(\epsilon)}{\log |Z|}. \tag{7}$$

**Proof:** First, let's define an indicator variable $E$ for the error event: $E = \begin{cases} 0, & \text{if } \hat{Z} = Z \\ 1, & \text{if } \hat{Z} \neq Z \end{cases}$. We have

$$\begin{aligned}
H(Z|\theta) &= H(Z|\hat{Z}, \theta) = H(Z, E|\hat{Z}, \theta) \\
&= H(E|\hat{Z}, \theta) + H(Z|E, \hat{Z}, \theta) \\
&= H(P_e) + P_e H(Z|E = 1, \hat{Z}, \theta) + (1 - P_e)H(Z|E = 0, \hat{Z}, \theta) \\
&\leq H(\epsilon) + P_e \log |Z|.
\end{aligned}$$

Hence, we have

$$P_e \geq \frac{H(Z|\theta) - H(\epsilon)}{\log |Z|} = \frac{H(Z) - I(Z;\theta) - H(\epsilon)}{\log |Z|}. \tag{8}$$

This part of the proof follows from Fano's inequality.

Now, since $I(Z;\theta) + I(R;\theta|Z) = I(Z, R;\theta)$, we have

$$\begin{aligned}
I(Z;\theta) &= I(Z, R;\theta) - I(R;\theta|Z) \\
&= I(R;\theta) + I(Z;\theta|R) - I(R;\theta|Z) \\
&\leq I(R;\theta) + I(Z;\theta|R)
\end{aligned}$$

Plugging in (8) obtains the result.

Note: $I(Z;\theta|R) = 0$ if $\theta$ is trained only on $R$, i.e., $Z$ and $\theta$ are conditionally independent given $R$.

**Lemma B.5.** *Let $\hat{Z}$ be the estimate of $Z$ based on $\theta$. Let $P_e = P(Z \neq \hat{Z}|\hat{Z}, \theta)$ be the probability of error in predicting $Z$ using $\theta$. Assume that $P_e \leq 0.5$. Then, we have that:*

$$P_e \leq \frac{H(Z) - I(R;\theta) + H(R|Z)}{\log 2}. \tag{9}$$

**Proof:** By definition of $E$ from Lemma B.4, we have

$$\begin{aligned}
H(Z|\theta) = H(Z|\hat{Z}, \theta) &= H(Z, E|\hat{Z}, \theta) \\
&= H(E|\hat{Z}, \theta) + H(Z|E, \hat{Z}, \theta) \\
&\geq H(E|\hat{Z}, \theta) \\
&= P_e \log(1/P_e) + (1 - P_e) \log(1/(1 - P_e)) \\
&\geq P_e \log 2
\end{aligned}$$

Hence, we have $P_e \leq H(Z|\theta) / \log 2$. Since $H(Z|\theta) = H(Z) - I(Z;\theta)$, and that

$$\begin{aligned}
I(Z;\theta) &= I(Z, R;\theta) - I(R;\theta|Z) \\
&= I(R;\theta) + I(Z;\theta|R) - I(R;\theta|Z) \\
&\geq I(R;\theta) - H(R|Z) + H(R|\theta, Z) \\
&\geq I(R;\theta) - H(R|Z),
\end{aligned}$$

the bound follows.

**Partial justification of Condition 2**: Consider the Markov chain $Z \to R \to \theta$ with additional arrows of $Z \to \theta$ and $R \to \theta$, where $Z$ is the preference, $R$ is the rationale, and $\theta$ is a trained model used to predict $Z$. From both Lemma B.4 and B.5, we see that as long as $\theta$ captures the information of $R$, i.e., $I(R;\theta)$ is large, and $R$ does not contain excessive irrelevant information other than $Z$, i.e., $H(R|Z)$ is small, the prediction error of $Z$ from model $\theta$ can be well-controlled. Specifically, based on the lower and upper bound analysis, we can conclude that the probability of error $P_e$ decreases with increasing $I(R;\theta)$ or decreasing $H(R|Z)$. This implies that the model does not need to capture additional information from $S$ to achieve high prediction accuracy for $Z$, i.e., $I(\theta_{ra}; S|Z, R)$ is small. In other words, the incorporation of $R$ in the training of $\theta_{ra}$ guides the model to easily predict $Z$ without resorting to finding potentially irrelevant information from $S$. A full justification of the condition hinges on a detailed analysis of the specific algorithm and is beyond the scope of this study.

### B.4 Proof of Theorem B.2 and Derivation of Regimes

Recall the definition of $\alpha, \beta, \epsilon, f(\cdot)$ in Sec. B.1. Now, we derive the relationship between the mutual information $I(Z; R|S)$ and these parameters:

$$
\begin{aligned}
P(Z = 1|S, R = 1) &= \frac{P(R = 1|Z = 1, S)P(Z = 1|S)}{P(R = 1|S)} \\
&= \frac{P(R = 1|Z = 1, S)P(Z = 1|S)}{\sum_{z \in \{0,1\}} P(R = 1|Z = z, S)P(Z = z|S)} \\
&= \frac{\beta(f(S) + \epsilon)}{\beta(f(S) + \epsilon) + \alpha(1 - f(S) - \epsilon)}.
\end{aligned}
$$

Similarly, we have

$$
P(Z = 1|S, R = 0) = \frac{\alpha(f(S) + \epsilon)}{\alpha(f(S) + \epsilon) + \beta(1 - f(S) - \epsilon)}.
$$

These equations show that the probability of $Z = 1$ given the query $X$, the preferred response $Y_1$, the dispreferred response $Y_2$, and the rationale $R$ depends on both the informativeness of the rationale, through $\alpha$ and $\beta$, and the informativeness of the query and responses, through $f(S)$. Using the above equations, we get the conditional entropies as follows:

$$
H(Z|S, R = 1) = H\left(\frac{\beta(f(S) + \epsilon)}{\beta(f(S) + \epsilon) + \alpha(1 - f(S) - \epsilon)}\right),
$$

$$
H(Z|S, R = 0) = H\left(\frac{\alpha(f(S) + \epsilon)}{\alpha(f(S) + \epsilon) + \beta(1 - f(S) - \epsilon)}\right)
$$

and substituting to the mutual information equation, we get:

$$
I(Z; R|S) = H(Z|S) - \sum_{r = \{0,1\}} P(R = r|S)H(Z|S, R = r).
$$

Then, we compute each probabilities $P(R|S)$ as follows:

$$
\begin{aligned}
P(R = 1|S) &= \sum_{z \in \{0,1\}} P(R = 1|Z = z, S)P(Z = z|S) \\
&= \beta(f(S) + \epsilon) + \alpha(1 - f(S) - \epsilon), \\
P(R = 0|S) &= 1 - P(R = 1|S) \\
&= 1 - (\beta(f(S) + \epsilon) + \alpha(1 - f(S) - \epsilon))
\end{aligned}
$$

and substitute them back into the conditional mutual information term, where we define $p = f(S)$:

$$
\begin{aligned}
I(Z; R|S) = H(p + \epsilon) &- (\beta(p + \epsilon) + \alpha(1 - p - \epsilon))H\left(\frac{\beta(p + \epsilon)}{\beta(p + \epsilon) + \alpha(1 - p - \epsilon)}\right) \\
&- (1 - (\beta(p + \epsilon) + \alpha(1 - p - \epsilon)))H\left(\frac{\alpha(p + \epsilon)}{\alpha(p + \epsilon) + \beta(1 - p - \epsilon)}\right).
\end{aligned}
$$

To study the influence of the parameters $\alpha, \beta, \epsilon$ on the mutual information, we consider the following edge cases:

**Regime 1: Highly informative rationale $\beta \approx 1$ and low noise $\alpha \approx 0 \implies$ Rationale is a sufficient statistics.**
In this case, the conditional probabilities are simplified to

$$
P(Z = 1|S, R = 1) \approx \frac{f(S) + \epsilon}{f(S) + \epsilon} = 1,
$$

$$
P(Z = 1|S, R = 0) \approx \frac{0}{1 - f(S) - \epsilon} = 0.
$$

Thus, the mutual information becomes as follows:

$$I(Z; R|S) \approx H(f(S) + \epsilon) - P(R = 1|S)H(1)$$
$$-P(R = 0|S)H(0) = H(f(S) + \epsilon).$$

In this regime, the conditional mutual information is solely determined by the entropy of the preference prediction, $H(f(S) + \epsilon)$. We notice that the entropy function is concave and reaches the max value at the 0.5 mark.

**Regime 2: Uninformative rationale $\beta \approx 0.5$ and high noise $\alpha \approx 0.5$.**

For this case, the conditional probabilities become:

$$P(Z = 1|S, R = 1) \approx \frac{f(S) + \epsilon}{f(S) + \epsilon + 1 - f(S) - \epsilon} = f(S) + \epsilon = P(Z = 1|S, R = 0)$$

and the mutual information equals to:

$$I(Z; R|S) \approx H(f(S) + \epsilon) - H(f(S) + \epsilon) = 0,$$

which shows that the rationales provides 0 information about the preference given the prompt and responses.

**Regime 3: Moderately informative rationale $\beta = 0.5 + \gamma$ and lower noise $\alpha = 0.5 - \gamma$.**

Given the assumption of $\beta = 0.5 + \gamma$ and $\alpha = 0.5 - \gamma$, where $0 \leq \gamma \leq 0.5$ and $\gamma$ denotes the level of informativeness of the rationale, we substitute this into the conditional mutual information term.

We first substitute into the conditional probabilities and get:

$$P(Z = 1|S, R = 1) = \frac{(0.5 + \gamma)(f(S) + \epsilon)}{(0.5 + \gamma)(f(S) + \epsilon) + (0.5 - \gamma)(1 - f(S) - \epsilon)},$$
$$P(Z = 1|S, R = 0) = \frac{(0.5 - \gamma)(f(S) + \epsilon)}{(0.5 - \gamma)(f(S) + \epsilon) + (0.5 + \gamma)(1 - f(S) - \epsilon)}.$$

Then, we compute the following probabilities:

$$P(R = 1|S) = \sum_{z \in \{0,1\}} P(R = 1|Z = z, S)P(Z = z|S)$$
$$= 0.5 + 2\gamma(f(S) + \epsilon - 0.5),$$
$$P(R = 0|S) = 0.5 - 2\gamma(f(S) + \epsilon - 0.5).$$

Now, we can compute the conditional mutual information term:

$$I(Z;R|S) = H(f(S) + \epsilon)$$
$$- (0.5 + 2\gamma(f(S) + \epsilon - 0.5)) \cdot H\left(\frac{(0.5 + \gamma)(f(S) + \epsilon)}{(0.5 + \gamma)(f(S) + \epsilon) + (0.5 - \gamma)(1 - f(S) - \epsilon)}\right)$$
$$- (0.5 - 2\gamma(f(S) + \epsilon - 0.5)) \cdot H\left(\frac{(0.5 - \gamma)(f(S) + \epsilon)}{(0.5 - \gamma)(f(S) + \epsilon) + (0.5 + \gamma)(1 - f(S) - \epsilon)}\right).$$
$$(10)$$

We can now analyze the behavior of the mutual information as a function of $\gamma$:

**When the rationale is uninformative $\gamma = 0$**, then the mutual information becomes 0, $I(Z; R|S) = 0$, which is consistent with previous cases, in which uninformative rationales provide no additional information about the preference $Z$ as demonstrated in Figure 4.

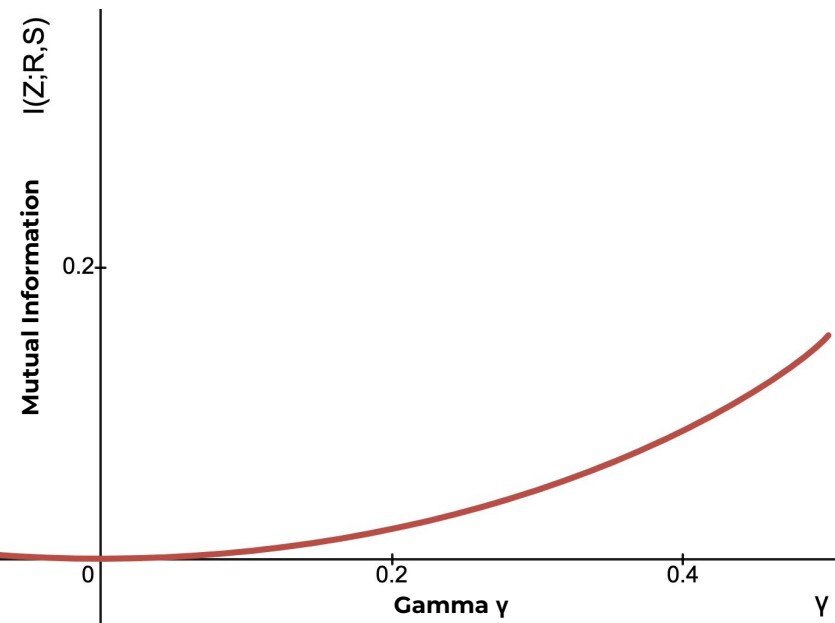

Figure 4: The plot of Equation 10 showing the relation between mutual information and gamma $\gamma$ for a fixed $f(S)$.

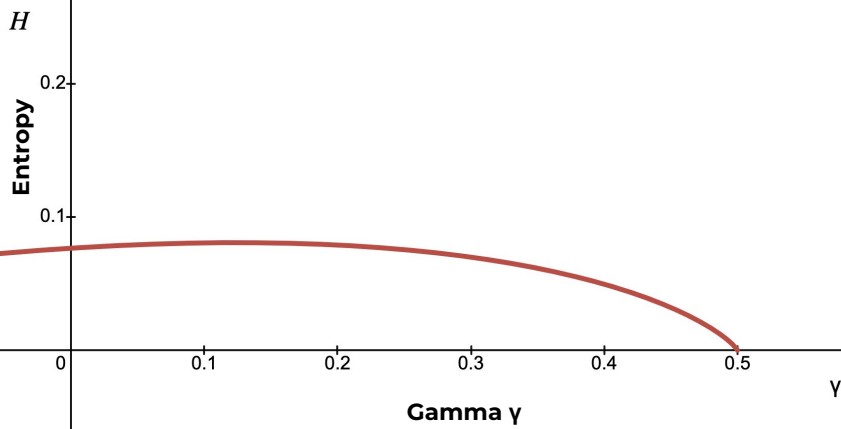

Figure 5: The plot of Equation 10 showing the relation between the first entropy term and gamma $\gamma$ for a fixed $f(S)$.

As rationale becomes more informative about the preference by increasing $\gamma$, we observe that mutual information also increases displayed in Figure 4.

Consider the case that the true probability $P(Z = 1|S) = f(S) + \epsilon > 0.5$, so the preference $Z$ is most likely to be 1.

Then, we now focus on terms in $I(Z; R|S)$ that contain $\gamma$. As $\gamma$ increases, the first entropy weight term $(0.5 + 2\gamma(f(S) + \epsilon - 0.5))$ increases and the second entropy weight term $(0.5 - 2\gamma(f(S) + \epsilon - 0.5))$ decreases. Entropy terms also involve $\gamma$, and we observe that as $\gamma$ increase, the ratio $\frac{(0.5+\gamma)(f(S)+\epsilon)}{(0.5+\gamma)(f(S)+\epsilon)+(0.5-\gamma)(1-f(S)-\epsilon)}$ approaches 1, since the numerator grows faster than the denominator. Thus, the first entropy term decreases with $\gamma$, as the entropy of a distribution to a deterministic one is lower.

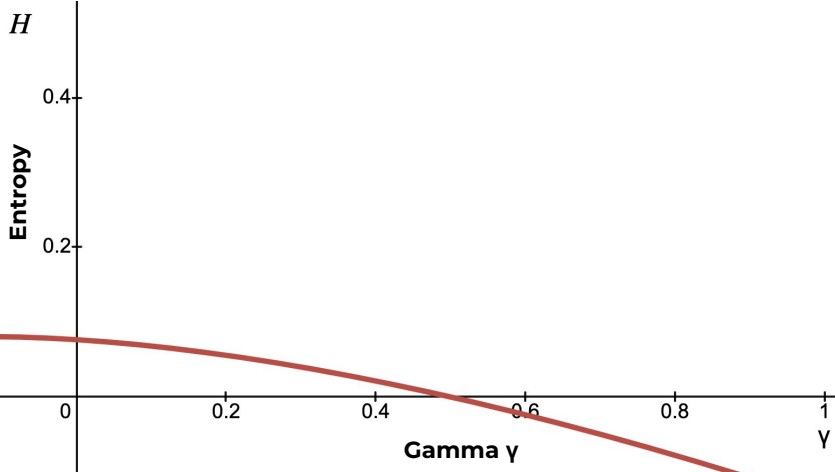

Figure 6: The plot of Equation 10 showing the relation between second entropy term and gamma $\gamma$ for a fixed $f(S)$.

Conversely, for the second entropy term, with the increase of $\gamma$, the ratio $\frac{(0.5-\gamma)(f(S)+\epsilon)}{(0.5-\gamma)(f(S)+\epsilon)+(0.5+\gamma)(1-f(S)-\epsilon)}$ approaches 0, so the second entropy term also decreases with $\gamma$.

- For the first entropy term, with an increase of $\gamma$, the weight of the first entropy term increases, but the entropy decreases itself (see Figure 5).
- For the second entropy term, with an increase of $\gamma$, the weight of the second entropy term decreases, and the entropy decreases itself (see Figure 6).

The net effect on the mutual information depends on the relative magnitudes of these changes. However, we can argue that the decrease in the entropy terms dominates the change in their weights due to the entropy function changing more rapidly near the extremes (i.e., when the distribution is close to being deterministic) compared to the middle range.

Thus, with an increase in $\gamma$, the overall contribution of the entropy terms to the mutual information decreases, causing an increase in $I(Z; R|S)$, which indicates that as the rationale becomes more informative about the preference, the mutual information increases.

## C    Additional Experimental Results

### C.1    Experimental Details

For DPO-based methods, we fine-tune the base model by supervised fine-tuning (SFT) with the chosen responses from the preference dataset for a single epoch. For ORPO, which avoids the reference model, we skip this SFT step. For models trained on RDPO and results reported in Section 4, we use $\gamma = 2.0$, and for RORPO, we use $\gamma = 10.0$. Similar to baseline methods, we train RDPO and RORPO for 1 epoch. We perform ablation studies on the hyperparameter $\gamma$ and the number of epochs in the following sections.

For our winrate scores, we report the mean winrates after querying the evaluator 3 times. To reduce the evaluator's order bias, we have additionally shuffled the order of responses. We note that the winrate error bars are within $< 3\%$ for 512 samples.

### C.2    Ablation Study on Models and Hyperparameters

|  | Mistral-7B-v0.1 | | | | | |
| --- | --- | --- | --- | --- | --- | --- |
| General | 62 | 61 | 63 | 61 | 62 | 60 |
| Detailed | 64 | 66 | 63 | 61 | 60 | 62 |
| $\gamma$ | 1.0 | 1.5 | 2.0 | 2.5 | 3.0 | 10.0 |

|  | Mistral-7B-v0.2-Instruct | | | | | |
| --- | --- | --- | --- | --- | --- | --- |
| General | 55 | 62 | 57 | 55 | 57 | 56 |
| Detailed | 56 | 56 | 57 | 60 | 57 | 59 |
| $\gamma$ | 1.0 | 1.5 | 2.0 | 2.5 | 3.0 | 10.0 |

|  | Zephyr-7B-Beta | | | | | |
| --- | --- | --- | --- | --- | --- | --- |
| General | 58 | 55 | 55 | 57 | 55 | 57 |
| Detailed | 48 | 49 | 51 | 51 | 50 | 51 |
| $\gamma$ | 1.0 | 1.5 | 2.0 | 2.5 | 3.0 | 10.0 |

Table 7: The impact of different values of hyperparameter $\gamma$ on the winrate of the RDPO model against the DPO model. The results on various models: Mistral-7B-v0.1 **(Top)**, Mistral-7B-Instruct-v0.2 **(Middle)**, and Zephyr-7B-Beta **(Bottom)**.

Here, we investigate the impact of the hyperparameter $\gamma$, ranging from 1.0 to 10.0, on the performance of the model trained with RDPO loss. We provide the winrate scores against the DPO model on the Orca dataset. As we see in Table 7, models trained on either general or detailed rationales can still achieve a stronger winrate against the DPO model, except for the case of *Zephyr-7B-Beta* model, which achieves a draw with the DPO model. For this model, the better quality rationales are important for effective preference learning, as the general rationales can still improve the performance.

### C.3    Ablation on the Number of Epochs

|  | General | Detailed |
| --- | --- | --- |
| Epoch 1 | 76 | 74 |
| Epoch 2 | 75 | 71 |
| Epoch 3 | 74 | 74 |

Table 8: The analysis of the number of epochs on the performance. Winrate of the RDPO model trained on the general rationales **(left)** and detailed rationales **(right)** against the DPO model, respectively.

In the main paper, we trained the models with the RDPO loss on a single epoch similar to DPO (Rafailov et al., 2024). Here, we study the impact of training the models with the rationales on more epochs. As we observe in Table 8, training with more epochs does not improve the winrate of the RDPO model against the DPO model. The reason could be that the model has already learned the preference well after the first epoch, or it could be that the quality of rationales could be further improved to increase the efficiency of the rationale-based preference learning algorithms.

## C.4 Rationale Quality Analysis

In this section, we examine the importance of the quality of rationales for preference learning. We study different types of rationales and possible errors encountered in rationales, and how these affect the preference learning of the model.

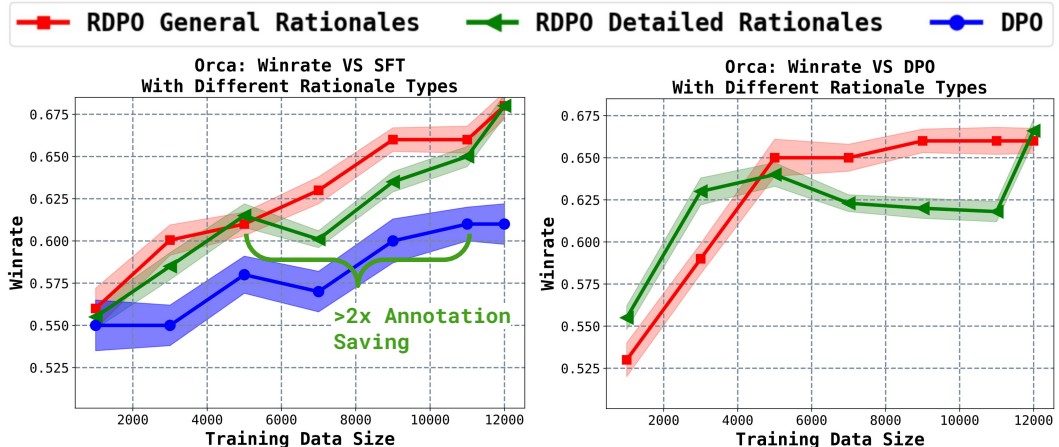

Figure 7: Measuring the impact of types of rationales on the RDPO performance. **Left:** Comparing the winrates against the SFT model. **Right:** Comparing the winrates against the DPO model.

**Detailed Rationales vs General Rationales.** Explaining why one answer is preferred over the other can be expressed in multiple ways through many perspectives. Here, we study the level of granularity of the rationales, general (which explains the preference at a high level without going into details) and detailed (which explains in details and pinpoints specifically to the prompt and the response). We use language models to automatically generate the rationales for the Orca dataset according to our intent. For details on generation prompts, we refer to Appendix C.9. We provide samples of these rationales in Appendix C.10. After training the models on respective rationales, we compare the winrates between RDPO trained models and the DPO one. Figure 7 on the left shows that the model trained on general rationales with the RDPO loss converges earlier to a high winning rate against the SFT model than the model trained on the detailed rationales. The reason could be that the general rationales share common features across the samples (e.g., clarity, conciseness, directness), which lets the model learn quickly and transfer these learning cues to other samples more easily, while detailed rationales might require more time to fully comprehend them. However, in both cases, the models trained on these rationales reach better winrates than the DPO against the SFT model. In Figure 7 (right), both RDPO models can have a better winrate > 57% against the DPO model with as few as 3,000 training samples, while the DPO model is trained on 11,000 samples. We provide results on models trained on additional epochs in Appendix C.3.

**Low-Quality Rationales.** RDPO has shown efficacy with rationales generated by the off-the-shelf models, even when the models have not undergone preference alignment. However, we want to further analyze the impact of rationale's quality on RDPO's performance. In particular, we examine the rationale quality in terms of its relevance and

| Permuted Rationales VS Original Rationales | | Opposite Rationales VS Original Rationales | |
|---|---|---|---|
| 1 | 99 | 10 | 87 |

Table 9: The analysis of quality of rationales on the RDPO performance. The winrate comparison between the RDPO models trained on rationales with errors and original rationales. **Left:** Permuted, irrelevant rationales. **Right:** Opposite, inaccurate rationales.

correctness. One case of a low-quality rationale can be a completely irrelevant rationale to the given pair of responses. To simulate irrelevant rationales, we permute the above-mentioned detailed rationales over different samples so that no rationale is relevant to the context. Training the model on these rationales with RDPO and comparing one trained on original rationales, we show in Table 9 that it achieves less than 1% winrate against the RDPO model trained on correct and relevant rationales. To study the impact of correctness, we negate the general rationales to have the opposite meaning and observe that the RDPO model trained on original rationales gets almost a 90% winrate. As we note, the quality of rationales is important for improving the preference learning performance. While we showed that rationales generated by the off-the-shelf language models can already bring significant improvement to preference learning, we expect that more deliberate control of the rationale quality can further improve preference learning. We leave the in-depth exploration of strategies for generating quality-controlled rationales to future work.

**Quality of Generated Rationales.** Here, we provide the quality of generated rationales judged by the GPT-4o in terms of helpfulness, relatedness, correctness, and coherence. As we observe in Table 10, the rationales generated are of good quality. We also shuffled the rationales to random prompts and also changed the meaning of rationales for sanity check of our evaluator and we observe significant drops in the scores. Please see C.12 for prompting details.

| | Helpfulness | Relatedness | Correctness | Coherence |
|---|---|---|---|---|
| Generated Rationales | 4.99 | 4.99 | 4.99 | 5.00 |
| Shuffled Rationales | 3.22 | 1.98 | 2.20 | 4.2 |
| Changed Meaning Rationales | 2.14 | 2.63 | 1.58 | 3.2 |

Table 10: The quality scores for generated rationales from the LLaMA-3.1-8B-Instruct model for UltraFeedback dataset. The scores are in the range from 1-5.

## C.5 Additional Results

We provide comprehensive results on AlpacaEval 2.0, Arena-Hard 0.1, and MT-Bench for the models corresponding to the Table 4 below:

| | LC Winrate | Winrate | Length | Arena-Hard 0.1 | MT-Bench |
|---|---|---|---|---|---|
| Original | 17.1 | 14.7 | 1676 | 12.6 | 7.6 |
| DPO | 19.5 | 15.8 | 1632 | 15.0 | 7.7 |
| Rationale-Only | 20.8 | 16.2 | 1630 | 15.2 | 7.5 |
| RDPO-General | 22.4 | 17.9 | 1627 | 16.2 | 7.6 |
| RDPO-Detailed | 23.2 | 18.6 | 1636 | 16.4 | 7.8 |

Table 11: Additional Benchmark Results

## C.6 Response Comparison

We compare the responses generated by the DPO- and RDPO-trained models. As shown in Table 12, the average output length by the DPO trained model is much longer than the RDPO

|  | Avg Output Length | | TriviaQA (Exact Match) | |
|---|---|---|---|---|
|  | DPO | RDPO | DPO | RDPO |
| Orca | 2021 | 1364 | 34.9 | 35.7 |
| UltraFeedback | 2066 | 1299 | 31.5 | 33.1 |

Table 12: Comparison between DPO and RDPO. **Left:** The average output lengths of the generated responses on the prompts from the test sets of respective datasets. **Right:** The exact match (EM) performance on the TriviaQA dataset of the preference-trained models on respective datasets.

trained model in the case of the Orca dataset, which is more than 60% longer on average. Due to longer output, there might be a chance for a higher occurrence of hallucinations. Therefore, we want to study the correctness of the outputs from these models. For this reason, we use the TriviaQA (Joshi et al., 2017) dataset and using LM Evaluation Harness (Gao et al., 2023) to measure the exact match (EM) accuracy and compare between the models. We see in Table 12 that models trained with the DPO loss experience a decrease in performance, compared to the models with RDPO loss. As a reason, we emphasize the importance of measuring the hallucinations in the generations of both models in future studies. We provide a comparison of the responses from the two models in Appendix C.13 and a time cost analysis in Appendix C.7 to help model owners better understand the trade-offs of our method.

### C.7 Log Probabilities.

To further investigate how rationales may boost preference optimization, we take a deeper look into log probabilities of the preference pairs. Specifically, we take a look at the difference of log probabilities between the chosen and rejected response on RDPO trained Llama-3.1-8B-Instruct model with Orca dataset.

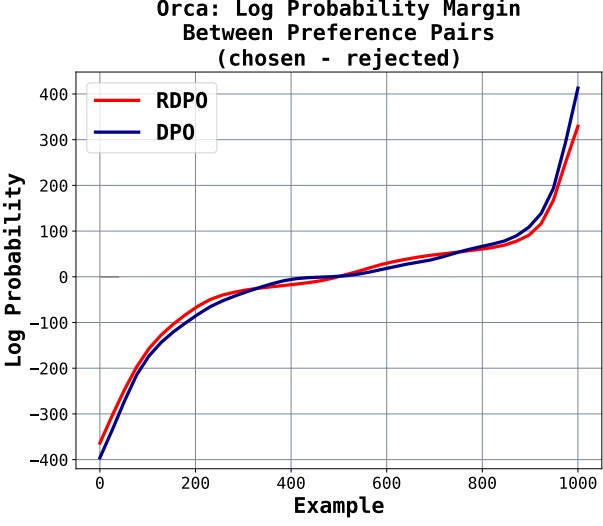

Figure 8: The log-probability margin between preferred and rejected responses. A comparison of the log probability margin of responses of a standard DPO-trained model against our Rationale DPO model. Y-axis denotes the margin as margin as logP(preferred response) − logP(rejected response). X-axis represents 1000 individual examples from the training set, which have been sorted in ascending order according to their calculated log-probability margin.

As shown in Figure 8, at the tails of the plots, where the margins are large, DPO yields higher margins than RDPO. This suggests that DPO may be exploiting certain biases or engaging in reward hacking to maximize reward margins. In contrast, RDPO appears to act as a

regularizer, mitigating such exploitative behavior. Interestingly, at the 0 level—where the margins are negligible, indicating that preference pair responses are difficult to differentiate, DPO remains relatively flat, whereas RDPO crosses the 0 line at a steeper angle. This suggests that RDPO is more effective at handling and resolving ambiguous cases compared to DPO. The investigation into the impact of rationales on preference learning suggests that they may serve as valuable cues, potentially acting as a regularizer to enhance preference learning.

### C.8 Cost Analysis

In this section, we analyze the cost breakdown to assist project owners in evaluating trade-offs. Specifically, we present a cost-benefit analysis of the approach. The table below outlines the cost of using the API to generate rationales for a given number of annotations. It also highlights the RDPO win rate compared to the SFT model for each data budget and estimates the number of annotations potentially saved by using RDPO to achieve the same level of performance as DPO.

While open-weight models were used to generate the rationales in our study, the table illustrates the associated costs of utilizing an API model, specifically gpt-4o-mini. The pricing for this model is $0.150 per 1M input tokens and $0.600 per 1M output tokens. The results shown in Table 6 are based on the Mistral-7B-v0.2-Instruct model trained on the Orca dataset.

We also report the runtime for RDPO and DPO for one epoch on the Llama-3.1-8B-Instruct model using 12,000 Orca examples, as follows:

- RDPO General: 6770 seconds
- RDPO Specific: 6950 seconds
- DPO: 3583 seconds.

While processing additional tokens nearly doubles the runtime, RDPO compensates for this by requiring fewer annotations while achieving comparable or superior performance to DPO. Furthermore, we report the average response lengths for the Orca dataset, highlighting that rationales are approximately 50% shorter in length compared to chosen or rejected responses:

- Chosen responses: 786
- Rejected responses: 981
- Rationale responses: 411

### C.9 Rationale Generation

Here, we provide the prompts to generate detailed and general rationales as mentioned in Section 4. In Prompt 1, we ask the model to generate general rationales without going into details and provide high-level reasoning. In Prompt 2, we ask the model to provide a specific rationale and go as much into detail as possible. These are our examples of prompts and we call for further studies to improve the prompts for higher quality rationales.

```
Given the prompt:
<prompt>
Which of the following assistant's responses is preferred and strictly follows
the prompt question? Why this assistant's response:
<chosen>
is preferred over the response provided below:
<rejected>
Why the response above is preferred over the response below? Provide a specific,
detailed explanation for your reasoning pointing directly at the response.
```

Prompt 2: Prompt to the LLM to retrieve a detailed rationale.

## C.10  Comparison of Rationales

We provide examples of the general and detailed rationales for the Orca dataset by the Mistral-7B-Instruct-v0.2, and compare them in Examples 1, 2, and 3. As we observe, the general and detailed rationales follow our intent. Additionally, we observe that a part of the general rationale is also included in the detailed rationale. It will be important to study, which type of rationale is more helpful for the preference training.

```
<General Rationale>
The first response strictly adheres to the prompt by providing the RDF
triplets in the required format, while the second response includes an
explanation that goes beyond the required format and may not be necessary for
the task at hand. Thus, the first response is preferred as it focuses solely
on providing the required output in the correct format.

<Detailed Rationale>
The assistant's response is preferred because it follows the standard format
for RDF triplets, where the subject comes first, followed by the predicate,
and then the object.  In the given input sentence, 'AFC Ajax (amateurs)'
and 'Ajax Youth Academy' are the subjects, 'has ground' and 'plays at' are
the predicates, and 'Sportpark De Toekomst' is the object that is common to
both subjects. Therefore, the RDF triplets should be structured as [subject,
predicate, object], with each triplet representing a distinct relationship.
The assistant's response correctly reflects this structure and semantics,
making it a preferred option.
```

Example 1: An example #1 of general and detailed rationales.

```
<General Rationale>
The preferred response strictly follows the prompt question by calculating
the total amount paid per episode for both major and minor characters
based on the given information.  It uses the provided information to
calculate the salaries of the major and minor characters and then adds them
together to find the total amount paid per episode.  The response below
includes unnecessary details and makes the response longer and less efficient.

<Detailed Rationale>
The assistant's response above is clear and accurate. They first calculate
the total amount paid to the minor characters per episode, which is $60,000.
Then, they calculate the total amount paid to the major characters per
episode, which is $225,000.  Finally, they add the salaries of the major
and minor characters to find the total amount James pays per episode, which
is $285,000." This explanation directly points to the steps taken in the
assistant's response and explains each step in detail.  It also uses clear
and concise language to make the explanation easy to understand.
```

Example 2: An example #2 of general and detailed rationales.

```
<General Rationale>
The response above is preferred because it directly answers the question
by specifically identifying the kidneys as the organs that secrete hormones
to help maintain homeostasis, while the response below goes into excessive
detail and provides additional information that is not directly related to
the question.

<Detailed Rationale>
The assistant's response is preferred because it provides a clear and
detailed explanation of the hormones secreted by the kidneys and their roles
in maintaining homeostasis.  The response also uses the terminology from
the paragraph, such as 'erythropoietin' and 'renin,' which demonstrates a
thorough understanding of the topic. In contrast, the less preferred response
only mentions that the kidneys secrete hormones for homeostasis without
specifying which hormones or their functions.  Therefore, the preferred
response is more informative and accurate, making it a better answer for the
given question.
```

Example 3: An example #3 of general and detailed rationales.

winrate using HALOs repository Ethayarajh et al. (2023)

## C.11 Evaluating Responses with LLM as a Judge

```
Here is a conversation between <user> and <assistant>.
<history>
<assistant> can now choose from different responses.
Response 1: <response1>
END OF RESPONSE 1
Response 2: <response2>
END OF RESPONSE 2
Between these two responses, which response is more helpful, harmless, and
concise? Pick one.
```

Prompt 3: Prompt to the LLM (e.g., GPT-4o) to retrieve the preference over two responses.

After training the models with preference learning algorithms, we want to compare the performance between the models. Thus, a common practice is to generate the responses from respective models and compare the quality between them. As a human evaluator might be expensive to acquire, we usually use a capable language model to judge the responses. In our case, we leverage the GPT-4o model to judge and provide winrate scores. We provide the prompt to determine a preferred response between generations in Prompt 3, which is implemented in the HALOs code implementation (Ethayarajh et al., 2023) and shuffles responses to avoid the ordering bias and averaged the scores over the runs.

## C.12 Evaluating Generated Rationale with LLM as a Judge

```
 Evaluate the quality of the rationale comparing the preference pairs for a
given prompt.
Provide the score through three categories of helpfulness, correctness, and
coherence with a range from 1 to 5, where:

Helpfulness:
5: Rationale provides significant insights and valuable guidance.
4: Rationale is useful and relevant, contributing positively to understanding.
3: Rationale offers some relevant information, but limited usefulness.
2: Rationale provides minimal assistance, possibly tangential.
1: Rationale is unhelpful, irrelevant, or misleading.

Relatedness:
5: Rationale fully relates and correctly answers the initial question.
4:  Rationale  relates  correctly  to  the  initial  question  with  minor
discrepancies.
3: Rationale relates correctly to the initial question with many discrepancies.
2: Rationale relates to the initial question but is tangential.
1: Rationale does not relate to the initial question.

Correctness:
5: Rationale is factually accurate and logically sound.
4: Rationale is accurate with minor inaccuracies.
3: Rationale contains a mix of correct and incorrect information.
2: Rationale contains significant inaccuracies or flawed logic.
1: Rationale is entirely false or fundamentally flawed.

Coherence:
5: Rationale is clear, well-organized, and easy to follow.
4: Rationale is generally clear and organized, with minor lapses.
3: Rationale is understandable, but lacks clarity in parts.
2: Rationale is difficult to follow due to poor organization.
1: Rationale is incomprehensible and disorganized.

Prompt:
<beginning of prompt>
%s
<end of prompt>

Rationale:
<beginning of rationale>
%s
<end of rationale>

Present the scores in a JSON dictionary format: scores = {"helpfulness": #,
"relatedness": #, "correctness": #, "coherence": #}.
```

Prompt 4: Prompt to the LLM (e.g., GPT-4o) to retrieve the score of a rationale.

Before training, we evaluate the generated rationales with a more capable model to check the quality of the generation. Please see the Prompt 4 for details. After verification, we drop the samples with low-scored rationales (any metric with a score 3 or less). Then, the new resulting filtered dataset, we use for further training.

### C.13   DPO and RDPO Generated Responses

We provide some examples and compare responses generated by the models trained with DPO and RPDO losses, respectively. In Example 5, we observe the case when DPO demonstrates the verbosity and generates a lengthy output compared to the response by the RDPO model. The length bias in DPO is also demonstrated in Example 4, which generates not only a longer output than RDPO but also outputs a personal opinion on a political stance. Example 6 asks for generating a tweet for a single sentiment. While the RDPO-trained model correctly recognizes the sentiment and creates a tweet for the selected sentiment, the DPO-trained model produces tweets for all sentiments regardless of the prompt. Despite the responses generated by RDPO being shorter on average than the ones by DPO, when prompted to generate an article, the RDPO response can be as lengthy as the DPO one as shown in Example 7.

```
<Prompt>
What is your opinion on Britain exiting the EU?
What is your opinion on Britain leaving the EU and Brexit?
Would you say that these questions are the same?

<DPO Response>
These questions are related but not exactly the same. Both ask for an opinion
on Britain's decision to leave the European Union (EU), which is known as
Brexit. However, the first question asks specifically about Britain exiting
the EU, while the second question expands to include the broader concept of
Brexit, which includes not only leaving the EU, but also the negotiations and
potential outcomes of that decision.

Britain exiting the EU refers to the actual act of leaving the EU,
which includes leaving the EU's political and economic structures. Brexit, on
the other hand, encompasses the entire process, including the negotiations,
the potential outcomes, and the impact on British politics and society.

So, if someone is asking for your opinion on Britain exiting the EU,
they are asking for your thoughts on the actual act of leaving the EU. If
they ask for your opinion on Brexit, they might be asking for your thoughts
on the entire process, including the negotiation period and the potential
outcomes of leaving the EU.

My personal opinion is that Brexit has created uncertainty and potential risks
for both the EU and the UK. However, it also gives the UK the opportunity to
set its own trade and regulatory policies, which could potentially lead to
new opportunities and economic growth.

<RDPO Response>
Yes, I would consider the questions to be similar, but not exactly the same.
The first question asks for your opinion on Britain's departure from the
EU, while the second question specifically refers to Brexit, which is the
name given to the process of Britain's departure from the EU. So, while the
second question is a subset of the first, it adds a layer of specificity by
referring to the name given to the process.
```

Example 4: A comparison of generated responses from DPO and RDPO models.

```
<Prompt>
I ' ve already decided that I want to live in Finland for a semester . What then ? What
if - I know , I ' m getting ahead of my own dream - world - he and I are together for so
long ... we fall in love ? What then ?
Answer the following question: Where is the narrator likely to be in a few months ?

<DPO Response>
In a few months, the narrator is likely to be living in Finland for a semester as they
have already decided on that.
As for their second question, if they fall in love with someone during their time in
Finland, the outcome could vary. The narrator could choose to continue their relationship
despite the distance if they are committed to making it work. They could also choose to
pursue the relationship further if both parties are interested.
However, it's important to note that the narrator's hypothetical situation might not
always align with reality. If they do fall in love and pursue a long-distance relationship,
it could require more effort, patience, and communication to maintain the relationship.
Ultimately, it depends on the individuals involved and their level of commitment to
making it work.
In any case, their current plan is to go to Finland for a semester, so that is where they
are likely to be in a few months.
Regarding the world reference in the second question, it's not clear how it relates to
the overall context of the scenario. If it refers to a fictional world, it's likely that
the question is asking about the potential outcomes of an imaginary situation.
However, if it refers to the real world, it's worth noting that long-distance
relationships can be challenging, but they are also possible with dedication, trust, and
open communication. If the narrator decides to pursue a relationship with someone, they
can consider using technology to stay connected, such as video calls, messaging, and
email. They could also set aside time for regular updates on each other's lives, and plan
visits to see each other as often as possible.
Ultimately, the key to maintaining a successful long-distance relationship is to approach
it with a positive attitude, patience, and an understanding that challenges will arise.
By staying committed to each other and working together to overcome any obstacles, the
narrator and their partner could potentially build a strong and lasting connection. So,
assuming the narrator's plan to go to Finland for a semester remains unchanged, they are
likely to be in Finland in a few months. Whether or not they fall in love, the outcome
depends on the individuals involved and their willingness to make the relationship work
despite the challenges.

                          Response continues...

<RDPO Response>
The narrator is likely to be living in Finland for a semester. However, if they fall in love
with someone, their plans might change depending on their feelings and circumstances.
```

Example 5: A comparison of generated responses from DPO and RDPO models.

```
<Prompt>
Sentiment possibilities. Choices:
(A). negative;
(B). positive;. Generate a tweet that has the following sentiment: (A).

<DPO Response>
(A). negative: "I just received a faulty product from the company, and it's
really frustrating! #CustomerServiceFail"
(B). positive: "Had an amazing time at the concert last night! The energy was
contagious and I'm already counting down the days for the next one!

<RDPO Response>
Here's a tweet that conveys a negative sentiment:
"Feeling down today. The rain outside matches my mood.
```

Example 6: A comparison of generated responses from DPO and RDPO models.

```
<Prompt>
Write an article based on this summary:
Jack Tector and Nathan Smith have received their first Ireland call-up for this month's
ICC Intercontinental Cup game against the Netherlands.

<DPO Response>
Title: Fresh Blood in the Irish Squad: Jack Tector and Nathan Smith Await Their Debut
against the Netherlands

Ireland, a long-standing cricket force in European cricket, is set to feature two
of its exciting young talents, Jack Tector and Nathan Smith, in this month's ICC
Intercontinental Cup game against the Netherlands. These young guns will be making their
debut on the international stage, providing them with an unforgettable experience and an
opportunity to shine amongst the cricketing world's elite.

Jack Tector, a 23-year-old all-rounder, has been a consistent performer in both
domestic and provincial cricket, earning him a well-deserved Ireland call-up. With his
skills in both batting and bowling, Tector offers immense potential to the side and can
add valuable depth to the Irish squad. His aggressive approach at the crease, coupled
with his ability to generate pace and bounce from the ball, make him a formidable prospect
to look out for.

Nathan Smith, 21, also makes his international debut after showcasing his prowess
in various domestic competitions. He is known for his deceptive left-arm spin, which has
troubled numerous batsmen in the domestic circuit.

<RDPO Response>
Ireland Calls Up Jack Tector and Nathan Smith for Upcoming ICC Intercontinental Cup Game
against Netherlands

The Irish cricket team has made some exciting announcements recently, as two young and
promising players, Jack Tector and Nathan Smith, have been granted their first call-ups
to the national squad for the ICC Intercontinental Cup game against the Netherlands. This
upcoming match is set to take place this month, and both Tector and Smith are eager to
make the most of this incredible opportunity.

Jack Tector, a 21-year-old all-rounder from Wiltshire, England, has been on the
radar of Irish selectors for some time now. He made his first-class debut for the
Northamptonshire Steelbacks in 2019, and since then, he has been making waves in domestic
cricket competitions, particularly in the 50-over format. Tector has been consistently
impressive with the bat, having scored a century and several half-centuries in various
domestic games. He is also a handy left-arm spinner, which adds depth to Ireland's
bowling attack.

Nathan Smith, on the other hand, is a 23-year-old left-handed opening batsman
from County Antrim, Northern Ireland.
```

Example 7: A comparison of generated responses from DPO and RDPO models.

