# OpenReview forum: "Data-Centric Human Preference with Rationales for Direct Preference Alignment"
_colmweb.org/COLM/2025/Conference — COLM 2025_

### Official Review · Reviewer_wt4m · 2025-04-30

**Rating:** 6
**Confidence:** 4
**Ethics Flag:** 1

**Summary:**

This paper suggests augmenting preference data with machine-generated rationales. The intuition is that the added explanations to justify the choices made by the humans help the model to better learn during the alignment. Their experiments show good win rates for the  Rationale-enriched preferences when using various optimizations techniques (DPO, ORPO, SimPO), and an impressive data efficiency.
The intuition is clear, as the results of the experiments, but the theoretical justification is less convincing –it's a pity that most of it is in the appendix– where it focuses on the added value of the rationales, without looking at its nature, i.e. being machine generated.
Despite the good results and the gained efficiency, there is a a troubling concern, at the epistemic level, in the "enrichment" of human preferences with model-generated "explanations".

**Reasons To Accept:**

The approach, both efficient, intuitive and versatile, shows good results:
- good data efficiency: the reduction of needed training samples is impressive.
- simplicity and generalization of the method: it seems indeed easy to improve the performance of existing techniques, e.g. DPO.
- empirical gains shown in the experiments.
- attempt at a theoretical backing, but not without caveat.
- versatility, as the method can be integrated with many preference learning algorithms

**Reasons To Reject:**

Alas, it's at least troubling to rely on a post-hoc machine generated rationales:
- epistemic risk: the rationales are machine-generated introducing a kind of circularity in the training.
- false sense of alignment: with such rationales and the reliance of an LLM-as-a-judge, aren't we being fooled here?
- limited theoretical backing: 1/ most of it is alas delegated to the appendix ; 2/ it relies on the rationale being what it claims to be when it has been generated —the authors say, page 22, that the rationale should "not contain excessive irrelevant information" but how is this guaranteed here...

---

> ### Author Response · Authors · 2025-06-03
>
> > 1. Despite the good results and the gained efficiency, there is a a troubling concern, at the epistemic level, in the "enrichment" of human preferences with model-generated "explanations". epistemic risk: the rationales are machine-generated introducing a kind of circularity in the training.
>
> We thank the reviewer for highlighting the potential epistemic risk of circular evidence. We want to clarify that our experimental setup rigorously mitigates this concern. Firstly, **all rationales are generated once, offline, by a frozen model before preference learning begins**. This ensures that the rationales remain static and are not influenced by, or adapted to, the evolving model during training, thereby breaking any direct feedback loop. Secondly, if a true circularity were at play, one would expect a significant performance degradation when rationales from an entirely different, unrelated generator are introduced. However, as detailed in Table 8, replacing our original rationale generator with a model of a different architecture and training corpus results in only a marginal change (at most 1–3 percentage points) in winrate. This empirically demonstrates that **the observed performance gains are robust and not attributable to shared biases between the rationale generator and the learned model**. Finally, these **rationales serve an auxiliary role**. Their purpose is to help annotators articulate complex human judgments and enable the model to internalize these nuances more effectively. **They do not override the human-provided preference ground truth labels**, which remain the primary drivers of the preference learning process. These deliberate design choices collectively bound the epistemic risk, while preserving the core contribution of our paper: enabling more efficient and effective preference alignment.
>
> ---
>
> > 2. false sense of alignment: with such rationales and the reliance of an LLM-as-a-judge, aren't we being fooled here?
>
> Thank you for raising this. We recognise the risk that an LLM could appear well-aligned simply because it can generate persuasive rationales that flatter another LLM acting as the judge. To minimise that risk we have verified **several safeguards**. Firstly, the **model that produces the rationales** is architecturally **distinct from** and never fine-tuned on the outputs of **the judging model** (GPT--4o). The generator therefore **could not overfit to the judge’s distribution**, reducing self-reinforcing bias. Additionally, we have **manually audited a random 5% of examples**, verifying whether the rationale is faithful to the underlying preference and whether the judge’s verdict matches human preference rankings (we provide some examples of our detailed and general rationales in Appendix C.8). Lastly, we injected counterfactual “misleading-but-eloquent” rationales and adversarially **tweaked rationales to see if the judge could be fooled**. As shown in Table 9 of Appendix, **failure cases (which is a response with an average score above 3) were rare (<3 %)**, suggesting the system is not merely rewarding rhetorical "smoothness". These checks give us confidence that rationales enhance genuine preference alignment rather than fooling us.
>
> ---
>
> > 3. limited theoretical backing: 1/ most of it is alas delegated to the appendix ; 2/ it relies on the rationale being what it claims to be when it has been generated —the authors say, page 22, that the rationale should "not contain excessive irrelevant information" but how is this guaranteed here…
>
> Due to strict page limits, the full proofs have been moved to Appendix B. This was a deliberate decision to ensure the main text remains focused on our central contribution: the **introduction of a low-cost method leveraging machine-generated rationales for enhanced preference learning**, supported by a solid empirical evaluation.
>
> Our theory aims to provide **intuitive guidance through a simplified model**, offering insights rather than formal worst-case guarantees. Similarly, the guideline stating that a rationale "should not contain excessive irrelevant information" should be interpreted as a **qualitative measure of practical utility, not an information-theoretic constraint**.
>
> **To empirically assess and ensure the quality of these generated rationales**, consistent with our guideline on avoiding excessive irrelevant information, we employed four distinct criteria (on a 1-5 scale): **helpfulness, relatedness, correctness, and coherence**. As detailed in Section C.4 of the Appendix, our evaluation demonstrates that the **machine-generated rationales achieve consistently high quality scores** (averaging almost 5), whereas intentionally **noisy rationales receive significantly lower scores** (below 3 on average), showing that LLM judges can identify and penalize potentially irrelevant information.

---

> > ### Comment · Reviewer_wt4m · 2025-06-06
> >
> > Thanks for your comments. It seems my main concern remains though: what you call "rationales" are post hoc generated explanations that fit the ground truth labels. The "circularity" I mentioned is that a model could generate valid looking explanations that differ from the true rationales of the human labelers —it's "circular" in the sense that the model learns from the generation of a model. It might give good results but it is also, at best, concerning, don't you think?

---

> > ### Author Response · Authors · 2025-06-08
> > **Thank you for your reply!**
> >
> > Thanks for your thoughtful comment. We agree that blindly training on a model’s own explanations could create a closed loop in which errors reinforce themselves.
> >
> > Our technique avoids that loop in **two ways**. First, we adopt a single frozen teacher to generate the rationales once and we do not run additional generation rounds. This “one-shot teacher-student” framework, which avoids recursive data regeneration [1], represents the standard approach in knowledge distillation and has demonstrated strong effectiveness [2,3]. Second, we apply a **quality filter** on the model-generated rationales. Each rationale is rated on correctness, helpfulness, relatedness, and coherence (details in Section C.4 and C.10). Rationales averaging < 3/5 are discarded from the original dataset. **This practice helps safeguard against low-quality data generation in self-training and distillation, where a (higher-capacity) teacher supplies high-quality outputs a single time to train a student [4, 6].**
> >
> > Finally, we note that existing reports of the risk of circularity, or “model collapse” (where model overfits to its own artifacts and its ability to generalize deteriorates) occur when models train over multiple rounds of generations without quality control [5, 7]. Our approach uses a single, filtered pass. The fact that it yields strong empirical gains suggests that data-quality filtering could neutralise this risk of training on model-generated data. While a comprehensive investigation of this direction is beyond the scope of this work, we will **include a brief discussion** to motivate future research on this topic.
> >
> > [1] Shumailov, Ilia, et al. "AI models collapse when trained on recursively generated data." Nature 631.8022 (2024): 755-759.
> >
> > [2] Hsieh, Cheng-Yu, et al. "Distilling Step-by-Step! Outperforming Larger Language Models with Less Training Data and Smaller Model Sizes." Findings of the Association for Computational Linguistics: ACL 2023. 2023.
> >
> > [3] Yang, Chuanpeng, et al. "Survey on knowledge distillation for large language models: methods, evaluation, and application." ACM Transactions on Intelligent Systems and Technology (2024).
> >
> > [4] Feng, Yunzhen, et al. "Beyond model collapse: Scaling up with synthesized data requires reinforcement." ICML 2024 Workshop on Theoretical Foundations of Foundation Models. 2024.
> >
> > [5] Gerstgrasser, Matthias, et al. "Is Model Collapse Inevitable? Breaking the Curse of Recursion by Accumulating Real and Synthetic Data." First Conference on Language Modeling.
> >
> > [6] Wu, Yue, et al. "Self-play preference optimization for language model alignment." arXiv preprint arXiv:2405.00675 (2024).
> >
> > [7] Dohmatob, Elvis, Yunzhen Feng, and Julia Kempe. "Model Collapse Demystified: The Case of Regression." The Thirty-eighth Annual Conference on Neural Information Processing Systems.

---

> > > ### Comment · Reviewer_wt4m · 2025-06-10
> > >
> > > your method suggests to augment human-labeled data with machine-generated texts that you call rationales. My main concern remains that they are not _actual_ rationales and that I'm not sure what the model is learning from them. Still, given your explanations I'm updating my rating.

---

### Official Review · Reviewer_gDSV · 2025-05-08

**Rating:** 7
**Confidence:** 4
**Ethics Flag:** 1

**Summary:**

## Summary of the paper
This paper proposes a modification of DPO-style preference tuning algorithms by augmenting them with explicit modeling of rationales. Concretely, instead of maximizing $\log p(y_w \succ y_l | x)$, the proposal is to maximize $\log p(y_w \succ y_l, r | x)$, where $y_w$, $y_l$, and $x$ are the winning response, losing response, and prompt respectively, and $r$ is a natural language rationale for why $y_w$ is better than $y_l$. The likelihood is decomposed into the original direct preference likelihood and $\log p(r | y_w \succ y_l, x)$. In practice, the rationales are generated from the policy model itself. The modification proposed is in principle applicable to any DPO-style algorithm, and the paper presents experiments with rationale-augmented versions of DPO, ORPO, and SimPO. The training datasets used are Orca DPO Pairs, binarized Ultrafeedback, and Anthropic HH, and the evaluation is 1) in-domain: measuring win rate of preference tuned models models on held out portions from the training datasets themselves; and 2) AlpacaEval 2.0 in a standard setting. The experiments show that rational augmentation improves all three algorithms on the in-domain evaluation, and DPO and ORPO on AlpacaEval (results from SimPO were not shown).

## Strengths
The proposed modification is interesting, and fairly cheap given that the policy model itself is used to generate the rationales. The implementation is sensible, and it is nice to see the empirical gains on three different algorithms and various datasets in the experiments presented.

## Weaknesses
1. Evaluation could be more thorough: Since the primary motivation of this work is to improve the generalizability of the algorithms, the paper would benefit from more experiments on OOD datasets, and tasks beyond AlpacaEval. For example, how does rationale augmentation impact the model's math (e.g., GSM8K, MATH), reasoning (e.g., BBH), and factuality (e.g., MMLU)? I recommend evaluating models on these additional benchmarks too.
2. While it is clear that obtaining rationales itself is fairly cheap, the rationale augmentation might increase the training time complexity significantly. It would be helpful to compare the base algorithms and their rationale augmented versions in terms on compute used as well.
3. The details of some the SimPO experiment in Section 4.1.1 and its purpose is unclear. The text refers to a "full size dataset", but it is unclear what this means. From the caption on Table 3, it looks like the experiment is comparing a RSimPO model trained on 20K samples against SimPO models trained on 20K and 40K instances. Is the point that RSimPO more sample efficient? Also the paragraph mentions filtering Ultrafeedback and Anthropic HH datasets, but it is not clear how and why they were filtered.

**Questions To Authors:**

1. How exactly is $p(r | y_w \succ y_l, x)$ computed in practice? It is clear that the model's assigned probability to target rationales is being computed, but what text is provided as context to the model?
2. It is not clear what the log-probabilities plot (Fig. 4) in Section 4.2 is showing. The caption seems to be referring to a different table. What does x-axis represent?
3. The text describing the SimPO experiment mentions filtering Ultrafeedback and Anthropic HH to make 40K sized datasets. What kind of filtering was done, and why?
4. The AlpacaEval experiments use only the Orca DPO Pairs dataset. Why was only this dataset chosen for these experiments, and not Ultrafeedback and Anthropic HH too?

**Reasons To Accept:**

The paper presents an interesting modification to preference optimization that is fairly general and easy to implement. The results show that the method may be effective at improving various DPO-style algorithms.

**Reasons To Reject:**

The experiments do not provide sufficient evidence that model developers should switch to using rationale-augmented versions of DPO style algorithms. There are some outstanding questions about the limitations of the proposal, e.g., the effect on various skills of the models.

---

> ### Author Response · Authors · 2025-06-03
> **Author Response (1/3)**
>
> > 1. Evaluation could be more thorough: Since the primary motivation of this work is to improve the generalizability of the algorithms, the paper would benefit from more experiments on OOD datasets, and tasks beyond AlpacaEval. For example, how does rationale augmentation impact the model's math (e.g., GSM8K, MATH), reasoning (e.g., BBH), and factuality (e.g., MMLU)?
>
> **Our work targets human preference optimization**, training LLMs to align with complex human value judgments, not factual or mathematical correctness.
> The datasets we chose cover open-ended, preference-driven tasks which test how well the model captures human preferences. Orthogonally, GSM8K benchmarks step-by-step logical reasoning and numeric accuracy, which are **valuable but outside our present scope**; adding it would not test preference alignment, but rather reasoning skills. Nevertheless, extending the rationale-guidance framework to math-focused tasks can provide a valuable test of its generality; we **plan to explore this in the follow-up work**.
> Instead, **we have added additional evaluations on OOD benchmarks on human preference such as AlpacaEval 2.0, Arena-Hard 0.1 and MT-Bench**. We are actively conducting these additional evaluations on MT-bench and Arena-hard and commit to incorporating these comprehensive results into our paper. The comprehensive results on AlpacaEval 2.0, Arena-Hard 0.1, and MT-Bench for the models corresponding to the Table 4 setup are presented below:
> |            	| LC Winrate | Winrate | Length | Arena-Hard 0.1 | MT-Bench |
> |----------------|------------|---------|--------|----------------|----------|
> | Original   	|   	17.1 |	14.7 |   1676 |       	12.6 |  	7.6 |
> | DPO        	|   	19.5 |	15.8 |   1632 |       	15.0 |  	7.7 |
> | Rationale-Only |   	20.8 |	16.2 |   1630 |       	15.2 |  	7.5 |
> | RDPO-General   |   	22.4 |	17.9 |   1627 |       	16.2 |  	7.6 |
> | RDPO-Detailed  |   	23.2 |	18.6 |   1636 |       	16.4 |  	7.8 |
>
> As the results demonstrate, both RDPO-General and RDPO-Detailed consistently outperform the DPO baseline across AlpacaEval 2.0 and Arena-Hard 0.1, confirming their **strong generalization capabilities beyond in-distribution data**. We will incorporate these comprehensive results into the revised version of our paper.
>
> ---
>
> > 2. While it is clear that obtaining rationales itself is fairly cheap, the rationale augmentation might increase the training time complexity significantly. It would be helpful to compare the base algorithms and their rationale augmented versions in terms on compute used as well.
>
> We appreciate the reviewer's concern regarding the computational overhead of rationale augmentation. We have addressed this point **in Appendix C.6, where we provide a comprehensive analysis of compute costs and training efficiency**, including direct comparisons with DPO methods. Our findings indicate that while the initial generation of rationales does incur a one-time cost, which aligns with the reviewer's assessment of it being "fairly cheap", this **investment yields a significant net positive**. The presence of high-quality rationales facilitates a **substantial reduction in the sample complexity** required for effective preference learning. This directly translates to considerable savings in total training time and compute resource utilization. For instance, as detailed in Table 11, our method can **reduce required annotations by up to a factor of 3** (saving approximately 6,000 annotations). Thus, **despite the per-sample overhead that might double processing time** for each sample, the overall computational efficiency of the training pipeline is significantly enhanced by drastically reducing data requirements.

---

> > ### Comment · Reviewer_gDSV · 2025-06-03
> >
> > Thanks for the additional evaluations and pointing me to the cost analysis in the appendix. They address weaknesses 1 and 2 in my list. I recommend moving the cost analysis to the main text.

---

> ### Author Response · Authors · 2025-06-03
> **Author Response (2/3)**
>
> > 3. The details of some the SimPO experiment in Section 4.1.1 and its purpose is unclear. The text refers to a "full size dataset", but it is unclear what this means. From the caption on Table 3, it looks like the experiment is comparing a RSimPO model trained on 20K samples against SimPO models trained on 20K and 40K instances. Is the point that RSimPO more sample efficient? Also the paragraph mentions filtering Ultrafeedback and Anthropic HH datasets, but it is not clear how and why they were filtered.
>
> We appreciate the reviewer's questions for clarification regarding the SimPO experiment in Section 4.1.1. The primary purpose of this experiment was two-fold: (1) To **demonstrate the scalability and performance improvement of RSimPO** when trained on a larger dataset. By "full-size dataset," we refer to the 40K instance version of the dataset, showcasing RSimPO's ability to **leverage more data for better performance**. (2) To **highlight RSimPO's annotation efficiency**: The comparison between RSimPO trained on 20K samples and the SimPO model trained on 40K instances directly illustrates that our method can achieve comparable performance with significantly less annotated preference data.
>
> Regarding the filtering of the Ultrafeedback and Anthropic HH datasets, we applied **two main criteria to ensure data quality**. Firstly, we **removed data points* where the preferred response was rated lower than the rejected response, as these indicate inconsistent or ambiguous human judgments. We **retained only instances** where the preferred response received a score of 5 or higher on a 10-point scale, ensuring that the 'preferred' choice genuinely represented a strong positive example. We appreciate the feedback and will incorporate this clarification directly into Section 4.1.1 and the relevant Appendix section of the revised manuscript. We will also **publish our complete data** upon the end of the review process.
>
> ---
>
> > Q1. How exactly is p(r|y_w > y_l, x) computed in practice? It is clear that the model's assigned probability to target rationales is being computed, but what text is provided as context to the model?
>
> We use the following prompt template to generate as well as to compute the rationale term p(r|x, y_w > y_l).
>
> *Prompt:*
>
> Given the prompt: Which of the following assistant’s responses is preferred and strictly follows the prompt question?
>
> Why this assistant’s response: is preferred over the response provided below:
>
> On a high and general level, why the response above is preferred over the response below?
>
> Provide a general, high-level explanation for your reasoning without going into the response’s details.
>
> *End of Prompt*
>
> We have provided all prompts in Section C.7 of our Appendix.
>
> ---
>
> > Q2. It is not clear what the log-probabilities plot (Fig. 4) in Section 4.2 is showing. The caption seems to be referring to a different table. What does x-axis represent?
>
> We apologize for the oversight regarding the misplaced caption for Figure 4 in Section 4.2. This will be corrected in the revised version of the paper. Figure 4 is designed to visually **represent the log-probability margin between preferred and rejected responses**, a key metric for evaluating preference learning. The y-axis quantifies this margin as log⁡P(preferred response)−log⁡P(rejected response). By plotting this difference, we can **assess how strongly a model distinguishes between preferred and rejected examples**.
> The plot specifically compares the log probability margin of responses of a standard DPO-trained model against our Rationale DPO model. The x-axis represents 1000 individual examples from our training set, which have been sorted in ascending order according to their calculated log-probability margin. This sorting allows for a clear visual comparison of how the model separates preferred responses over rejected ones.
>
> As illustrated in the figure, DPO yields margins that are both larger at the tails and smaller closer to the zero-margin point compared to RDPO. This pattern suggests **DPO's potential for exploiting certain biases or engaging in reward hacking**, while **RDPO appears to act as a regularizer**, mitigating such behavior and ultimately enhancing the overall preference learning process.

---

> > ### Comment · Reviewer_gDSV · 2025-06-03
> >
> > Thanks for the clarification, and for agreeing to update the paper accordingly. This addresses W3 in my list of weaknesses. I increased my score.

---

> > > ### Author Response · Authors · 2025-06-06
> > >
> > > Thank you for the prompt reply. We appreciate your thoughtful feedback and the time to update your assessment. We will update the paper accordingly.

---

> ### Author Response · Authors · 2025-06-03
> **Author Response (3/3)**
>
> > Q3. The text describing the SimPO experiment mentions filtering Ultrafeedback and Anthropic HH to make 40K sized datasets. What kind of filtering was done, and why?
>
> We applied **two primary criteria to ensure the high quality and reliability of the preference data** used for training. We **discarded data points** where the preferred response was rated lower than or equal to the rejected response. This step was crucial for eliminating inherently inconsistent or ambiguous human judgments. We **retained only instances** where the preferred response received a score of 5 or higher on a 10-point scale. This ensured that the 'preferred' choice genuinely represented a positive example. We **will integrate this clarification** into the details section of the revised manuscript.
>
> ---
> > Q4. The AlpacaEval experiments use only the Orca DPO Pairs dataset. Why was only this dataset chosen for these experiments, and not Ultrafeedback and Anthropic HH too?
>
> The Orca DPO Pairs dataset, **due to its manageable size, allowed us to conduct extensive hyperparameter sweeps and robust, iterative evaluations**. This was crucial for assessing core performance metrics without prohibitive resource expenditure and, critically, for thoroughly testing a **wide range of hyperparameters to identify the optimal settings** for each baseline and variant of our method. However, our core evaluation methodology across all datasets (Orca DPO, Ultrafeedback, and Anthropic HH) has **consistently relied on direct head-to-head model comparisons**. This approach was chosen to precisely **measure preference learning efficacy** and provide the most granular and direct insights into each method's performance. While AlpacaEval 2.0 is an important and widely recognized benchmark, it was incorporated **primarily for comprehensiveness and broader context**. The Orca DPO Pairs dataset allowed us to showcase the foundational benefits of our method in a setting where we could conduct the most thorough experiments. We will incorporate the comprehensive results into the revised version of our paper.

---

### Official Review · Reviewer_DP63 · 2025-05-12

**Rating:** 7
**Confidence:** 4
**Ethics Flag:** 1

**Summary:**

This paper proposes enriching human preference datasets with rationales which are explanations for why one response is preferred over another. Then, this can be used to improve Direct Preference Optimization (e.g., DPO). By providing deeper context and reducing ambiguity in preference data, models can better understand human values, leading to improved alignment, higher benchmark performance, and more efficient use of existing annotations. This represents a data-centric shift in alignment research, focusing on enriching rather than expanding datasets. The authors also conduct extensive experiments to show

**Questions To Authors:**

Please refer to the weakness part.

**Reasons To Accept:**

1. The paper is well written and easy to follow.

2. The problem of inserting rationale to enhance performance of direct alignment algorithms.

3. The experiments are reasonable and can verify the effectiveness of their approach.

**Reasons To Reject:**

1. Can authors give more details about how their method generate rationale to build the dataset with rationale, like prompt example?

2. The dataset used in this paper is limited. It's better for users to consider some other datasets like math related (GSM8k) which can be used to further verify the effectiveness of their approach.

3. The authors miss some analysis of hyperparameter $\gamma$ which is really important in this work.

---

> ### Author Response · Authors · 2025-06-03
>
> > 1. Can authors give more details about how their method generate rationale to build the dataset with rationale, like prompt example?
>
> We have provided all details for generating both general and detailed rationales in Section C.7 in Appendix.
> For generating the general rationales, we employ the following prompt template to create our dataset:
>
> *Prompt:*
>
> Which of the following assistant’s responses is preferred and strictly follows the prompt question? Why this assistant’s
>
> response: (chosen) is preferred over the response provided below: (rejected) On a high and general level, why the response above is
>
>  preferred over the response below? Provide a general, high-level explanation for your reasoning without going into the response’s details.
>
> *End of Prompt*
>
>
>
> We will move them to the main paper for improved clarity.
>
> ---
>
> > 2. The dataset used in this paper is limited. It's better for users to consider some other datasets like math related (GSM8k) which can be used to further verify the effectiveness of their approach.
>
> **Our work targets human preference optimization**, training LLMs to align with complex human value judgments, not factual or mathematical correctness.
> The datasets we chose cover open-ended, preference-driven tasks which test how well the model captures human preferences. Orthogonally, GSM8K benchmarks step-by-step logical reasoning and numeric accuracy, which are **valuable but outside our present scope**; adding it would not test preference alignment, but rather reasoning skills. Nevertheless, extending the rationale-guidance framework to math-focused tasks can provide a valuable test of its generality; we **plan to explore this in the follow-up work**.
>
> ---
>
> > 3. The authors miss some analysis of hyperparameter gamma which is really important in this work.
>
> We provide the ablation studies on the hyperparameter $\gamma$ in Section C.2 of our Appendix.

---

> > ### Comment · Reviewer_DP63 · 2025-06-05
> >
> > Thanks for your response! I think the author has resolved my questions and I will raise my score.

---

> > > ### Author Response · Authors · 2025-06-06
> > >
> > > Thank you for the update! We are happy we could address your questions and appreciate you updating the score!

---

### Official Review · Reviewer_8NKj · 2025-05-21

**Rating:** 6
**Confidence:** 4
**Ethics Flag:** 1

**Summary:**

This paper introduces a new data-centric perspective into LLM alignment for achieving better explainability and performance through rationale for preference. To be specific, unlike the previous approaches (e.g., DPO) only use the preference label, the proposed framework newly introduce rationale to support the preference label and the target model is trained to maximize the likelihood of such rationale along with the conventional alignment objective under preference label. These rationales are generated by the target model itself via prompting. The effectiveness of the proposed rationale-augmented alignment framework has been demonstrated with various experiments.

**Reasons To Accept:**

1. **Well-motivated problem.** The explainability of LLM preference is an interesting and important direction to align LLM in real-world.

**Reasons To Reject:**

1. **Limited evaluation.** Most of evaluations are conducted on in-distributed test dataset, except Table 2 on AlpacaEval 2.0. It would be nice if the authors can show the evaluation results on other common benchmarks in LLM alignment such as MT-bench [1] or Arena-hard [2]. In addition, in Table 4, the authors show that only training with rationale yield better alignment performance under in-distribution scenario (Orca train -> Orca eval). It would be nice if the author can show the evaluation results on this setup using common benchmark like AlpacaEval 2.0.

2. **Editorial comments.** Currently, the detailed explanation and experiments about rationale is located in Appendix; for example, how to construct them (line 1118) or how the different choice of generation model affect the performance (Table 8). As this information is one of the most important part, I recommend move this into main draft instead of Figure 4 and relevant discussion or Section 5.

3. **Typos.** There are many critical typos in the current draft.
- Line 314: Figure 2 -> Table 2
- Line 1122: Figure 8 -> Table 8

[1] Zheng et al., Judging LLM-as-a-Judge with MT-Bench and Chatbot Arena., NeurIPS 2023
[2] Li et al., From Crowdsourced Data to High-Quality Benchmarks: Arena-Hard and BenchBuilder Pipeline., arXiv:24.06

---

> ### Author Response · Authors · 2025-06-03
>
> > 1. Limited evaluation. Most of evaluations are conducted on in-distributed test dataset, except Table 2 on AlpacaEval 2.0. It would be nice if the authors can show the evaluation results on other common benchmarks in LLM alignment such as MT-bench [1] or Arena-hard [2]. In addition, in Table 4, the authors show that only training with rationale yield better alignment performance under in-distribution scenario (Orca train -> Orca eval). It would be nice if the author can show the evaluation results on this setup using common benchmark like AlpacaEval 2.0.
>
> We appreciate the reviewer’s suggestion to include benchmarks like MT-bench [1] and Arena-hard [2]. While our initial evaluation primarily focused on demonstrating head-to-head performance within the distribution of our training datasets and the efficiency of our proposed preference learning methods, we recognize the importance of broader evaluation.
> We are actively conducting these additional evaluations on MT-bench and Arena-hard and commit to incorporating these comprehensive results into our paper. The **comprehensive results on AlpacaEval 2.0, Arena-Hard 0.1, and MT-Bench for the models** corresponding to the Table 4 setup are presented below:
> |            	| LC Winrate | Winrate | Length | Arena-Hard 0.1 | MT-Bench |
> |----------------|------------|---------|--------|----------------|----------|
> | Original   	|   	17.1 |	14.7 |   1676 |       	12.6 |  	7.6 |
> | DPO        	|   	19.5 |	15.8 |   1632 |       	15.0 |  	7.7 |
> | Rationale-Only |   	20.8 |	16.2 |   1630 |       	15.2 |  	7.5 |
> | RDPO-General   |   	22.4 |	17.9 |   1627 |       	16.2 |  	7.6 |
> | RDPO-Detailed  |   	23.2 |	18.6 |   1636 |       	16.4 |  	7.8 |
>
> As the results demonstrate, both RDPO-General and RDPO-Detailed consistently outperform the DPO baseline across AlpacaEval 2.0 and Arena-Hard 0.1, confirming their **strong generalization capabilities beyond in-distribution data**. We will incorporate these comprehensive results into the revised version of our paper.
>
> ---
>
> > 2. Editorial comments. Currently, the detailed explanation and experiments about rationale is located in Appendix; for example, how to construct them (line 1118) or how the different choice of generation model affect the performance (Table 8). As this information is one of the most important part, I recommend move this into main draft instead of Figure 4 and relevant discussion or Section 5.
>
> Thank you for flagging the importance of those details. Our original layout choices were driven by the page limit. We will move this information from the Appendix (line 1118, and Table 8) into the main paper.
>
> ---
>
> > 3. Typos. There are many critical typos in the current draft. Line 314: Figure 2 -> Table 2,  Line 1122: Figure 8 -> Table 8
>
>
> We appreciate identifying these typos and we have fixed them.

---

> > ### Comment · Reviewer_8NKj · 2025-06-04
> >
> > I appreciate the authors' detailed rebuttal along with the new experiments to address my concerns. Now, my concerns are mostly resolved. After reading this and also checking other reviewers' comments and authors' rebuttal for them, I decide to keep my original rating, leaning to the acceptance.

---

> > > ### Author Response · Authors · 2025-06-06
> > >
> > > We are very glad to hear that we have been able to resolve most of your concerns. We appreciate your positive assessment of our work!

---

### Decision · Program_Chairs · 2025-07-08

**Decision:**

Accept

**Comment:**

The paper explores a data-centric approach to enhance LLM alignment, particularly direct preference alignment. To improve learning efficiency, they proposed to augment preference pairs with additional machine-generated rationales that explicitly explain why a response should be preferred over another. Experimental results validate the effectiveness of the approach achieving higher performance on in-domain evaluation as well as AlpacaEval.


**Strength:**

All reviewers unanimously appreciate the well-motivated problem, simple and effective approach, sound experimental design-- testing the approach across various algorithms and datasets. The empirical gains were consistent and convincing across different evaluation settings.

During rebuttal, the authors adequately addressed other minor concerns, providing additional results to show generalization, add clarification and methodological details. The authors are highly advised to incorporate suggestions and comments in the camera ready version to enhance readability and overall impact.